# CAF08 adjuvant enables single dose protection against respiratory syncytial virus infection in murine newborns

Simon D. van Haren 1,2,11✉, Gabriel K. Pedersen3,11, Azad Kumar4, Tracy J. Ruckwardt 4, Syed Moin4, Ian N. Moore5, Mahnaz Minai5, Mark Liu 1, Jensen Pak 1, Francesco Borriello1,2,6,10, Simon Doss-Gollin 1, Elisabeth M. S. Beijnen 1, Saima Ahmed7, Michaela Helmel7, Peter Andersen3,8, Barney S. Graham4, Hanno Steen 1,7, Dennis Christensen 3 & Ofer Levy 1,2,9

Respiratory syncytial virus is a leading cause of morbidity and mortality in children, due in part to their distinct immune system, characterized by impaired induction of Th 1 immunity. Here we show application of cationic adjuvant formulation CAF08, a liposomal vaccine formulation tailored to induce Th 1 immunity in early life via synergistic engagement of Toll-like Receptor 7/8 and the C-type lectin receptor Mincle. We apply quantitative phospho-proteomics to human dendritic cells and reveal a role for Protein Kinase C-δ for enhanced Th1 cytokine production in neonatal dendritic cells and identify signaling events resulting in antigen cross-presentation. In a murine in vivo model a single immunization at birth with CAF08-adjuvanted RSV pre-fusion antigen protects newborn mice from RSV infection by induction of antigen-specific CD8+ T-cells and Th1 cells. Overall, we describe a pediatric adjuvant formulation and characterize its mechanism of action providing a promising avenue for development of early life vaccines against RSV and other respiratory viral pathogens.

[1] Precision Vaccines Program, Division of Infectious Diseases, Boston Children's Hospital, Boston, MA, USA. [2] Department of Pediatrics, Harvard Medical School, Boston, MA, USA. [3] Center for Vaccine Research, Statens Serum Institut, Copenhagen, Denmark. [4] Vaccine Research Center, National Institute of Allergy and Infectious Diseases, National Institutes of Health, Bethesda, MD, USA. [5] Infectious Disease Pathogenesis Section, National Institute of Allergy and Infectious Diseases, National Institutes of Health, Bethesda, MD, USA. [6] Department of Translational Medical Sciences and Center for Basic and Clinical Immunology Research (CISI), University of Naples Federico II, Naples, Italy. [7] Department of Pathology, Boston Children's Hospital and Harvard Medical School, Boston, MA, USA. [8] Department of Immunology and Microbiology, University of Copenhagen, Copenhagen, Denmark. [9] Broad Institute of MIT and Harvard, Boston, MA, USA. [10] Present address: Generate Biomedicines, Cambridge, MA, USA. [11] These authors contributed equally: Simon D. van Haren, Gabriel K. Pedersen ✉email: simon.vanharen@childrens.harvard.edu

nfectious diseases are the leading global cause of death in children under 5 years of age[1]. Among the viral pathogens of early life is respiratory syncytial virus (RSV), an enveloped, non-segmented negative-strand RNA virus of the family Pneumoviridae that is a common viral cause of serious respiratory illness in children under 5 years of age. Indeed, RSV is estimated to cause 3.2 million hospitalizations and at least 59,000 deaths worldwide, predominantly in children under the age of 6 months[2] rendering it the second most likely single pathogen to cause death in children under 1 year of age[3]. RSV infection is a particular threat in the very young due to limited airway size as well distinct features of the infant innate immune system[4]. Specifically, the Th2-biased response observed in newborns is likely associated with more severe manifestations of RSV infection[5,6], as it correlates with airway eosinophilia[7]. Similarly, the propensity of newborns to produce CXCL8 leads to infiltration of neutrophils, which also contributes to disease exacerbation[8,9].

A key approach to protect infants from viral pathogens such as RSV is immunization. Unfortunately, although immunization of infants with a formalin-inactivated preparation of RSV (FI-RSV) induced antibody (Ab) responses, it led to enhanced disease progression and in some cases even death after subsequent RSV infection[10]. Lung and blood pathology of these immunized children, as well as mechanistic insight from murine studies, revealed that both antibody and cell-mediated immune responses associated with severe disease. Induction of Abs with poor neutralizing activity led to immune complex deposition and complement activation in small airways and Th2-biased T cell response coincided with excess infiltration of neutrophils and eosinophils into the lungs[11,12].

In addition to a predominant Th1 response, potential immunization against RSV would require the generation of Abs capable of neutralizing the virus. The RSV fusion (F) glycoprotein is the target of most RSV-neutralizing Abs, specifically in its metastable prefusion conformation (pre-F)[13]. Recent studies of the F protein in complex with Abs have led to structure-guided stabilization of a recombinant F antigen variant that retains its pre-F state[14,15]. This variant, also known as DS-Cav1, is a promising vaccine candidate, eliciting high affinity Abs with strong neutralizing ability in mice, non-human primates, and antigen-experienced humans[16,17]. While these new developments are encouraging, the pace of translation in this area has been slowed by hesitation related to the aforementioned history of the formalin-inactivated vaccine resulting in enhanced RSV disease in infants[18].

A key approach to enhance and shape immune responses in distinct populations is the use of adjuvants. This approach may be particularly important in newborns and infants as they often do not generate a sufficiently protective immune response when vaccinated with a single dose of vaccinal antigen at birth[19–21]. Studies modeling the function and phenotype of the immune system in neonates in vitro and ex vivo have demonstrated a distinct neonatal innate immune system that responds differently to pathogen-associated molecular patterns (PAMPs), as compared to young adults or elderly adults[22–27]. In this context, as many vaccine adjuvants are designed to enhance the immunostimulatory effect of vaccines via activation of pattern recognition receptors (PRRs)[28] recognizing such PAMPs, these adjuvants may not optimally induce protective immune responses through vaccination at birth. The most well studied PRRs in newborns are Toll-like receptors (TLRs), type I transmembrane glycoproteins expressed on the cell surface or intracellularly. Recognition of microbial products by TLRs is a central event in the host response to pathogens. As compared to their adult counterparts, hallmark features of TLR activation in newborn monocytes or dendritic cells (DCs) include reduced NF-κB signaling[29,30], reduced production of T-helper 1 (Th1)-polarizing cytokines TNF-α and IL-

12p70[31–34], and increased production of Th2/anti-inflammatory cytokines IL-6 and IL-10, and of neutrophil chemoattractant CXCL8[33,35–39]. Activation of human neonatal leukocytes via imidazoquinoline agonists of TLR7/8, however, can enhance the magnitude of the newborn immune response in vitro[40,41] through involvement of the NLRP3 inflammasome[40], and can dramatically accelerate and enhance immunogenicity of pneumococcal conjugate vaccine (PCV13) in newborn non-human primates[42]. However, studied in vitro, TLR7/8 agonists alone were not able to sufficiently change the type of monocyte-derived dendritic cell (MoDC) immune response to a predominantly Th1 phenotype that is likely needed for protection against intracellular pathogens such as RSV[43,44].

We have previously conducted a screen of TLR and CLR adjuvant combinations that demonstrated that newborn DCs can be activated to instruct a Th1 response by combined stimulation through TLR7/8 and the C-type Lectin Receptor (CLR) Mincle, using R848 and trehalose-6,6-dibehenate (TDB), respectively[43]. This combined adjuvantation system results in NF-κB activation, as well as activation of the NLRP3 inflammasome and induces production of TNF-α, IL-1β and IL-18, cytokines that contribute to Th1 polarization. Furthermore, production of Th2/anti-inflammatory cytokines IL-10 and IL-12p40, and of neutrophil chemoattractant CXCL8 was reduced relative to each of the components tested alone. As a result, these DCs induce the differentiation of newborn naïve CD4+ T cells into Th1 cells instead of Th2[43]. The highly synergistic and age-specific fashion in which the R848 + TDB combination acts on newborn DCs makes it suitable for potential development of an early life vaccine against RSV. Our prior studies also showed that formulation of liposomes with TDB enables efficient induction of germinal center formation[45] as well as polyfunctional T cells[46] in neonatal mice.

Herein, we evaluated the ability of liposomal formulations containing the lipidated TLR7/8 agonist 3M-052, a Mincle agonist and the RSV pre-F antigen, to induce a potent and balanced Th1/Th2 immune response in newborn mice. We also dissected the molecular mechanism of synergistic activation of newborn human DCs by R848 + TDB, using label-free quantitative phosphoproteomics. Phosphoproteomic analysis of DCs indicated that synergistic activation of cytokine signaling pathways in newborn DCs by R848 + TDB is driven by TDB-induced endocytosis and prolongation of phosphorylation of the δ-isoform of protein kinase C (PKC-δ). In addition, it revealed marked synergistic enhancement of antigen cross-presenting ability, which was confirmed in human CD8+ T cell activation studies in vitro as well as in newborn immunization model in vivo. Immunization of newborn mice demonstrated the ability of CAF08, a liposomal formulation consisting of combined TLR7- and Mincle-agonists and RSV pre-F, to induce strongly neutralizing antibodies of IgG2a/c isotype as well as pre-F -specific Th1 and CD8+ T cell responses and to protect against RSV infection. This adjuvant platform for neonatal vaccines represents a promising approach to enhance the efficacy of a future vaccine against RSV and other pediatric pathogens against whom robust Th1 immunity is required for protection.

## Results

**Phosphoproteomic analysis of human moDCs reveals a link between endocytosis and the magnitude of cytokine induction in newborn cells.** We have previously shown that combined in vitro stimulation of human newborn MoDCs with R848 and TDB synergistically enhances the production of Th1-polarizing cytokines, such as TNF-α, in an age-specific manner, such that the effect of this combination was only additive in infants/young adults and actually antagonistic in elderly adults[43]. Key roles for

NF-κB, Syk and the NLRP3 inflammasome in overcoming the intrinsic impairment of newborn cells to instruct Th1-differentiation of CD4+ T cells were identified[43]. However, the extent of synergistically induced TNF-α and the age-specific nature of this synergy imply that additional signaling events may occur. Therefore, we utilized a mass spectrometry-based method for label-free quantitation of phosphorylation events induced by stimulation of MoDCs. This method was optimized to identify and quantitate hundreds of phosphorylation events in only 3-5 million human primary cells, thus enabling application of this technique to human newborn blood-derived primary leukocytes. Between 250 and 900 phosphorylation events were identified and quantified per sample. Supplementary Datasets 1–6 list all identified phosphorylated peptides reported in this study, their relative abundance in each sample, and each analysis step that was taken, as also outlined in the flow diagram in Supplementary Fig. 1. In newborn ($n = 9$) and adult ($n = 4$) study participants, depending on the treatment and age group, 26 to 989 phosphorylation sites changed their phosphorylation status in a statistically significant manner (Supplementary Fig. 1, Supplementary Datasets 1–6). A reduction in phosphorylation, as compared to vehicle-treated controls, was seen for a proportion of phosphorylation events in newborn MoDCs, but not adult MoDCs. As a potential explanation for this phenomenon, we speculate that it is possible that MoDCs may display an elevated level of baseline phosphorylation of certain proteins as a result of C-section-induced activation. Phosphorylation events that were altered by treatment in a statistically significant manner were subsequently selected for pathway overrepresentation analysis using InnateDB.com. Pathway overrepresentation analysis, mining multiple protein function databases (KEGG[47], REACTOME[48], PID NCI[49], BioCarta[50], NetPath[51] and INOH[52]), for increased confidence, provided evidence for the synergistic induction of cytokine secretion signaling pathways by R848 + TDB in newborn MoDCs, as compared to stimulation with each individual agonist (Fig. 1a). This synergy is absent in adult MoDCs, where treatment with single agonists results in the greatest induction of phosphorylation events in cytokine signaling pathways (Fig. 1b). In addition, a marked synergy in phosphorylation in endocytosis-related signaling pathways was observed in newborn MoDCs, but not in adult MoDCs. Further analysis of individual proteins involved in these pathways revealed the significant phosphorylation of PKC-δ playing a role in the cytokine signaling pathways as well as the endocytosis pathways, providing a possible link between these two processes. As R848 induces activation through endosomal receptors (TLR7/8), we hypothesized that increased endocytosis could be the cause of the observed synergy in cytokine signaling induction. The activation of one phosphorylated protein that was categorized in multiple endocytosis and cytokine signaling pathways, PKC-δ (phosphorylated on serine 645), was previously identified as a key component of both endosomal trafficking and signaling[53,54]. It has also been identified as an important downstream signaling component of Syk and can induce phosphorylation of Signal transducer and activator of transcription 1 (STAT1) on serine 727[55,56]. Both phosphorylations, PKC-δ S645 and STAT1 S727, were synergistically induced in newborn moDCs by R848 + TDB, as demonstrated with phosphoproteomics (Fig. 1c, d), as well as by western blotting (Fig. 1e). Images in Fig. 1e are representative of three biological replicates. Changes in phosphorylation were quantified using imageJ[57] (Supplementary Fig. 2). Indeed, a correlation between endocytosis-induced PKC-δ phosphorylation and production of cytokines was observed after inhibition of PKC-δ S645 phosphorylation by clathrin inhibitors (Fig. 1g) and inhibition of TNF-α secretion after treatment with a PKC-δ inhibitor (Fig. 1f). In addition, inhibition of endocytosis with

clathrin inhibitors reduced (R848 + TDB)-induced TNF-α production in newborn MoDCs to R848-like levels (Fig. 1h), further indicating the role of endocytosis in enhancement of cytokine production in newborn DCs.

**R848 and TDB synergistically induce antigen cross-presentation in newborn and adult moDCs.** In addition to synergistic induction of phosphorylation in cytokine and endocytosis pathways in newborn MoDCs, R848 + TDB also synergistically induced the phosphorylation of nine proteins in antigen processing and cross-presentation in both adult and newborn MoDCs. In total, 17 phosphorylated proteins in the MHC I antigen processing and presentation pathway were identified by LC/MS, nine of which showed increased phosphorylation upon treatment with R848 + TDB (Fig. 2a–c). STRING analysis[58] illustrates the type and degree of interactions between the identified proteins within this pathway (Fig. 2a). Several of the proteins in which we identified phosphorylation upon R848 + TDB treatment (shown in bold) are known to physically interact with each other. The phosphorylation of four key proteins in the process of antigen cross-presentation, including Sec22b[59–61] and Syntaxin-4 (STX4)[59,62], was confirmed in adult MoDCs by western blotting after electrophoresis using phos-tag™ reagent to retard the migration of phosphorylated proteins (Fig. 2b). Significant R848 + TDB-induced phosphorylation of additional cross-presentation-related proteins are shown in Fig. 2c.

The functional implications of these findings were investigated using a MoDC:CD8 co-culture in which adult MoDCs were pulsed with a soluble protein antigen, to which our study participants had pre-determined pre-existing CD4- and CD8-T cell immunity. Contrary to a CD4+ T cell recall response, a CD8+ recall response in this assay would require cross-presentation of the internalized and processed protein on MHC I. Indeed, robust induction of a CD8+ recall response to any of the four proteins tested was observed only when MoDCs were also activated with R848 + TDB (Fig. 2d). These observations confirm the ability of combined R848 + TDB stimulation to synergistically induce antigen cross-presentation. Protein levels of a key signaling adapter identified in our phosphoproteomics platform, Sec22b[59,60], were diminished in DCs using a pool of 4 siRNAs (Supplementary Fig. 3). Reduction of Sec22b using siRNA, but not a pool of 4 non-targeting siRNAs (Mock siRNA), resulted in diminished cross-presentation of influenza hemagglutinin (HA) antigen, as R848 + TDB adjuvant combination no longer induced the activation of autologous HA-specific CD8+ T cells.

**Immunization at birth with the pre-F antigen in combination with liposomal adjuvant formulation incorporating TLR7/8 + Mincle agonist (CAF08) improves vaccine immunogenicity.** A range of imidazoquinoline congeners were formulated into the liposomal adjuvant CAF01 consisting of dimethyldioctadecylammonium (DDA) and TDB for evaluation of optimal incorporation efficacy and adjuvant stability (Supplementary Fig. 4). 3M-052, a lipidated congener of R848 with potent TLR7/8- and inflammasome-activating ability in vitro and in vivo[42,63,64], incorporated most efficiently. Liposomes containing both TDB and 3M-052 demonstrated stability in terms of size and surface charge over several weeks (Supplementary Fig. 4). We therefore decided to use this formulation for the in vivo studies in mice. Formulations with varying combinations of TDB and 3M-052 incorporated into DDA liposomes were therefore prepared and combined with 20 µg/ml RSV pre-F antigen[15,16] to evaluate the adaptive immune response induced by synergistic activation via TLR7/8 and Mincle in newborn and adult mice. Adult doses of 100 µL therefore consisted of 2 µg of pre-F antigen. Newborn

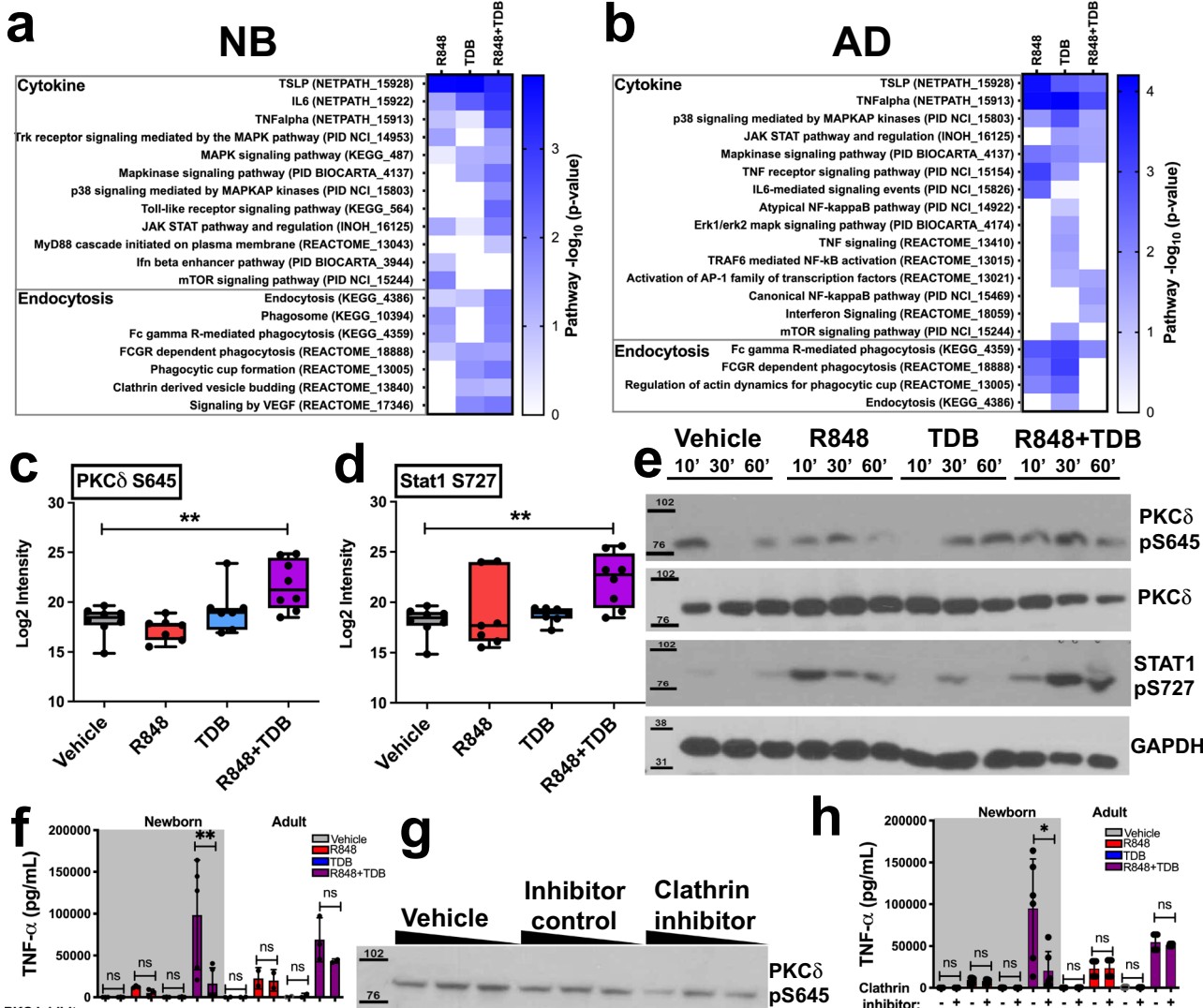

**Fig. 1 Phosphoproteomic analysis and use of targeted inhibitors in human MoDCs reveals a role for endocytosis in synergistic enhancement of cytokine production in newborns.** Newborn and adult MoDCs were stimulated with 50 μM R848, 100 ug/ml TDB, combination, or vehicle control for 30 min. **a**, **b** Pathway overrepresentation analysis (hypergeometric test with Benjamini Hochberg correction) of phosphopeptides quantified through mass spectrometry reveals synergistic enrichment of both cytokine- and endocytosis pathways by R848+TDB in newborn, but not adult, cells (hypergeometric test). Significant increase in phosphorylation (paired Student's $t$ test) in newborn MoDCs of PRKCD at Ser645 and its substrate, STAT1 at Ser727, (**c**, **d**) are shown using box-whisker plots (center:mean, boxes:75th and 25th percentile, whiskers: minimum and maximum values) was confirmed by western blotting (**e**), demonstrating prolonged phosphorylation of these two residues after stimulation with R848+TDB. **f**–**h** Inhibition of endocytosis with a clathrin inhibitor reduced PRKCD phosphorylation and subsequent TNF-α production by neonatal MoDCs. $n = 9$ newborns, 4 adults, error bars indicate mean +SEM. Statistical comparisons in (**c**, **d**, **f**, and **h**) are indicated by connecting lines and employed two-tailed Wilcoxon rank-sum test (*$p < 0.05$, **$p < 0.01$, ***$P < 0.001$). Source data are provided as a Source Data file.

mice were given half the adult dose. Immunization with DDA liposomes containing RSV pre-F and either 3M-052 (DDA052), or TDB (CAF01), or TDB and a low dose of 3M-052 (CAF08 low), or TDB and a high dose of 3M-052 (CAF08 high), as well as pre-F alone or formulated in an MF-59-like oil-in-water emulsion (Addavax) all elicited a high titer of pre-F -specific IgG antibodies in newborn as well as adult mice (Fig. 3). Newborn mice reached comparable levels of anti-pre-F antibodies to adults. Notably, in liposome formulations containing 3M-052, alone or in combination with TDB, the antibody response was skewed to IgG2a/c and less to IgG1, as compared to the other vaccine formulations. This is further illustrated in the graphs depicting the IgG2a/IgG1 ratio. This ratio was the highest in DDA052- and CAF08-immunized groups. In saline-immunized groups, this ratio should not be over-interpreted, as both the IgG1 and IgG2a titers were

within the lower regions of detectability. In mice, an antibody response skewed to IgG2a/c isotypes may indicate a Th1-driven immune response rather than a Th2-driven response[65], or, alternatively, a T cell-independent immune response[66]. Serum from immunized newborn mice was able to neutralize the infection of HEp-2 cells by RSV in vitro (Fig. 3). The highest neutralization titers were observed in groups that received pre-F formulated in either CAF01 or CAF08. Phenotypic analysis of draining lymph node B- and T cells confirmed the formation of B cell germinal centers upon immunization with pre-F-containing formulations (Supplementary Fig. 5). In accordance with the RSV-specific antibody titers in Fig. 3, these germinal centers contained predominantly IgG1+ B cells in pre-F only, CAF01 and Addavax groups, and B cells of alternate IgG isotypes in DDA052 and CAF08 groups. A significant increase in T follicular helper

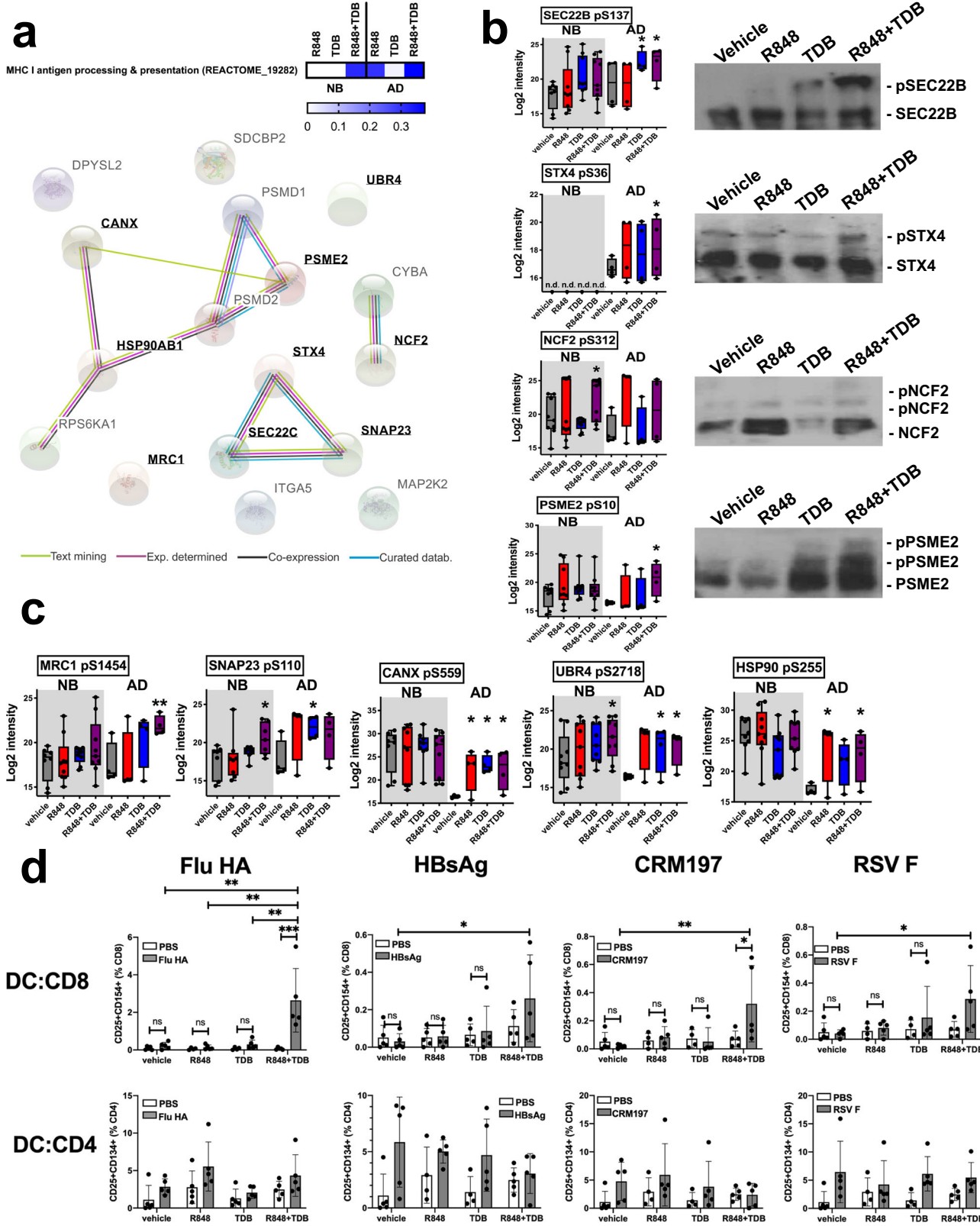

cells (Tfh) was also seen in DDA052, CAF01 and CAF08 groups, but only in adult mice. There was no apparent increase in Tfh cells in draining lymph nodes or spleens from newborn mice.

**Immunization with CAF08 induces RSV F-specific Th1 and CD8[+] T cells.** To evaluate RSV F-specific CD4[+] T cells, we developed an MHC II tetramer presenting RSV pre-F peptide 235-249 (REFSVNAGVTTPVST). The tetramer specificity was validated on lymph node cell suspensions from immunized CB6F1 mice. Staining of CD4[+] cells from newborn mice immunized with pre-F-containing formulations consistently stained positive (1–2%) with the RSV F235-249 tetramer, but not the CLIP tetramer (Fig. 4a, b). Cells from saline-injected mice did

**Fig. 2 Stimulation of human MoDCs in vitro with (R848+TDB) induced antigen cross-presentation of internalized proteins.** Phosphoproteomic analysis of human MoDCs identified the phosphorylation of 17 proteins belonging to the MHC class I processing and presentation pathway (Reactome 19282). **a** STRING analysis illustrates the order of interaction between these identified proteins. Proteins significantly phosphorylated upon treatment are indicated in bold. **b**, **c** The phosphorylation of nine of these proteins was significantly increased upon stimulation, shown as Box-whisker plots, center:mean, boxes:75th and 25th percentile, whiskers: minimum and maximum values). **b** The treatment-specific phosphorylation of four proteins was confirmed by western blotting after separation of phosphorylated and unphosphorylated species of each protein using phos-tag gel electrophoresis(representative images of three biological replicates). Cells from adult study participants were used for this. **c** Comparative intensity of the five remaining analytes with significant differences in phosphorylation $N = 9$ newborns, 4 adults, error bars indicate mean+SEM. Statistical comparisons in (**b**, **c**) are indicated by connecting lines and employed two-tailed Wilcoxon rank-sum test (*$p < 0.05$, **$p < 0.01$, ***$P < 0.001$). **d** Cross-presentation was confirmed in a MoDC:CD8 co-culture assay. MoDCs were pulsed with 1 µg/ml soluble protein antigen as indicated, in the presence or absence of adjuvants as indicated on the axes, and subsequently cultured with autologous CD4$^+$ or CD8$^+$ T cells for 4 days before evaluation of antigen-specific T cell activation ($n = 5$ adults, 2-way ANOVA with Tukey post-hoc test, error bars indicate mean+SEM. *$p < 0.05$, **$p < 0.01$, ***$P < 0.001$). Source data are provided as a Source Data file.

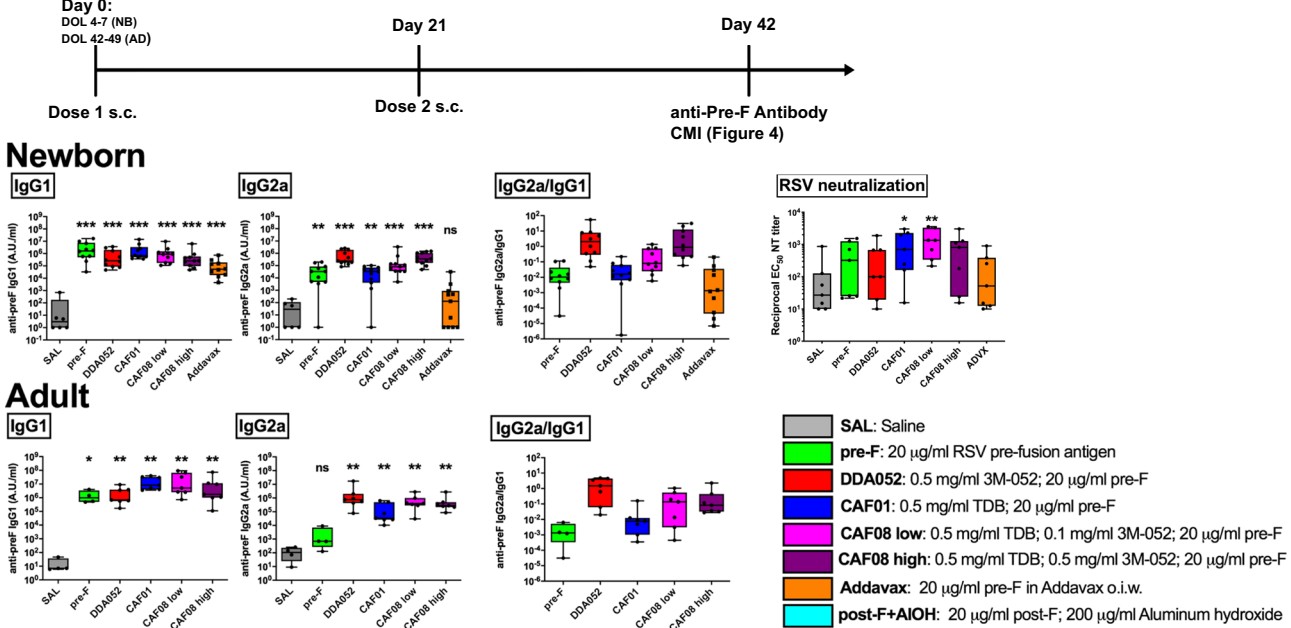

**Fig. 3 RSV pre-F in liposomes adjuvanted with 3M-052 with or without TDB-induced potent IgG2a/c antibody responses in newborn and adult mice.** Immunization of newborn and adult CB6F1 mice demonstrates potent induction of anti-pre-F antibodies. The switch to IgG2a/c isotype was greatest after immunization with liposomes containing 3M-052. Immunization of newborn CB6F1 mice shows potent induction of RSV-neutralizing antibodies in a HEp-2 cell neutralization assay. All treatment groups except the Addavax-treated group had mean neutralization titers ≥ 3-fold above the saline group. Statistical analysis employed the Mann–Whitney $U$ test comparing to the saline or pre-F groups ($n = 7$ adult mice/group, 10 newborn mice/group, shown as Box-whisker plots, center:mean, boxes:75th and 25th percentile, whiskers: minimum and maximum values. Statistical comparisons to SAL group employed two-tailed Mann–Whitney $U$ test (*$p < 0.05$, **$p < 0.01$, ***$P < 0.001$)). Source data are provided as a Source Data file.

not stain positive with the RSV F235-249 tetramer. In addition, single-cell sorting of tetramer-positive cells and clonal expansion with anti-CD3/CD28-coated microbeads (Thermo Fisher) generated a tetramer-positive CD4$^+$ T cell clone, further confirming tetramer staining specificity (Supplementary Fig. 6E). CD4$^+$ T cells from newborn mice in all treatment groups except the saline group exhibited similar frequencies of tetramer-positive CD4$^+$ T cells in the draining lymph node, but less in the spleen (Fig. 4b). The difference between treatment groups was evident when the relative percentage of T-bet expressing tetramer-positive cells (Th1) and GATA-3 expressing tetramer-positive cells (Th2) was determined (Fig. 4c). The relative ratio of antigen-specific Th1 cells over Th2 cells was significantly increased after immunization with CAF08 high only, although CAF01 and CAF08 low also showed non-significant trends towards enhanced Th1 responses. Furthermore, T-bet and GATA-3 staining on tetramer-negative cells clearly indicates that the induction of RSV-specific Th1 cells in newborn mice, which are intrinsically

skewed to a Th2 response, is selective and antigen-specific, as no change in T-helper ratios was observed in tetramer-negative cells (Supplementary Fig. 6). Ex vivo re-stimulation of cell suspensions derived from draining lymph nodes or spleens with pre-F antigen confirmed the presence of IFN-γ secreting CD4$^+$ Th1 cells in newborn mice immunized with CAF08, but not any of the other formulations (Fig. 4d). Analysis of intracellular cytokines by flow cytometry demonstrated a significant percentage of RSV F-specific Th1 cells, as well as CD8$^+$ T cells, induced only in newborn mice immunized with CAF08-adjuvanted vaccine. The finding of antigen-specific CD8$^+$ T cells after immunization with a vaccine adjuvanted with CAF08, which activates newborn cells through TLR7/8 and Mincle, further documents induction of antigen cross-presentation as noted in human MoDCs in vitro (Fig. 2). Analysis of secreted cytokines upon re-stimulation confirmed a higher preference for secretion of Th2/anti-inflammatory cytokines IL-5, IL-13 and IL-10 by cells from newborn mice as compared to adult mice (Fig. 4d). Secretion of IFN-γ was most

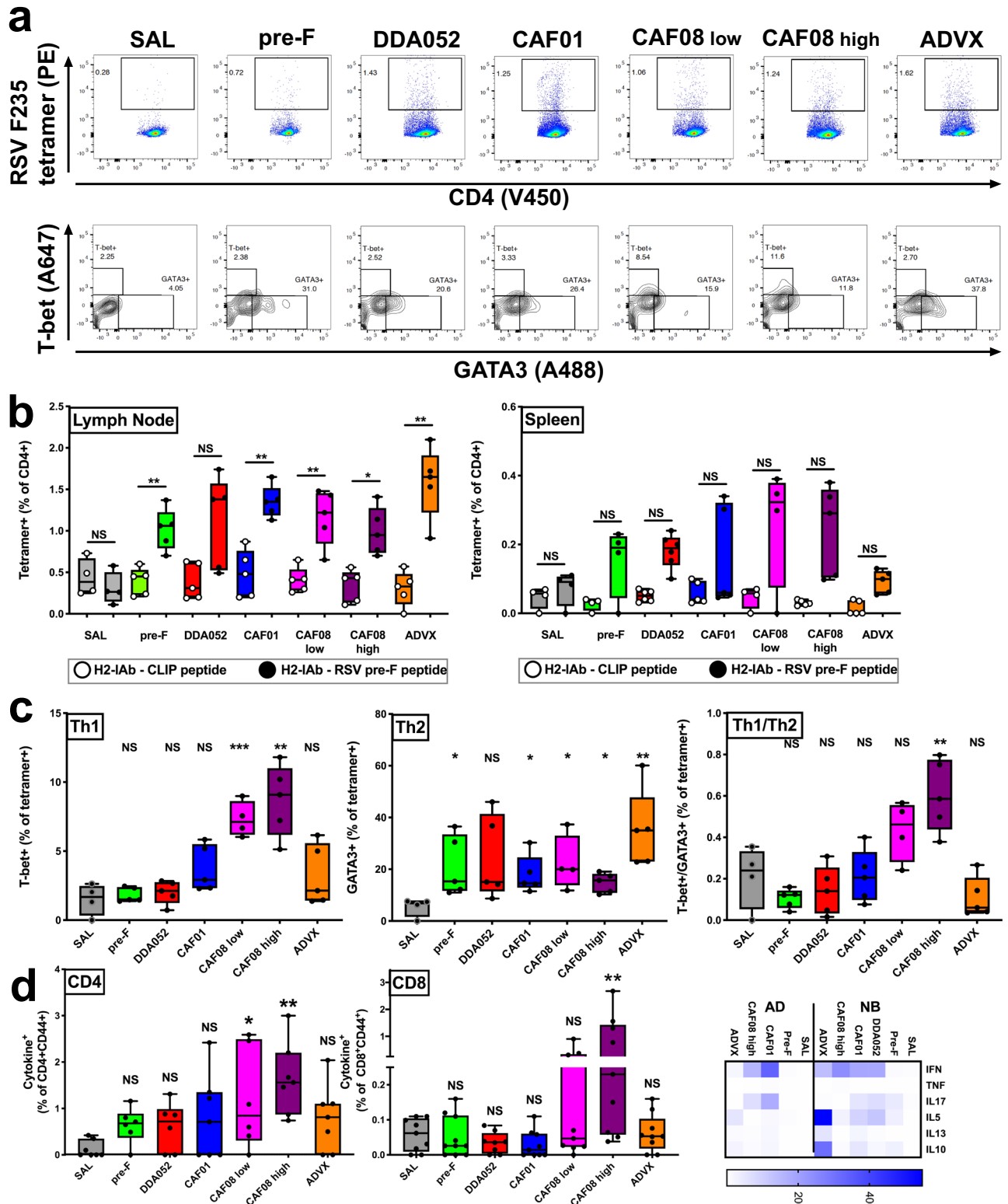

pronounced in adult mice after immunization with CAF01 or CAF08, whereas in newborn mice the greatest IFN-γ secretion was observed in mice immunized with CAF08, coinciding with a reduction in secretion of IL-5 and IL-10 as compared to other antigen-containing immunization groups. These observations confirm the ability of combined TLR7/8 and Mincle stimulation in newborn leukocytes to induce Th1 polarization and reduce the intrinsic neonatal Th2 bias[43].

**CAF08-adjuvanted RSV pre-F protein protects mice from RSV infection**. We compared the ability of Alum, CAF01, CAF08, or Addavax, when added to RSV pre-F, to enhance protection against RSV challenge in neonatal mice. Post-F adsorbed to Aluminum hydroxide (AlOH), was included as a control condition to emulate the Th2-driven immune stimulating ability of formalin-inactivated RSV. DDA052 was not a stable formulation and was hence excluded. Mice were immunized at birth with a single immunization and

**Fig. 4 CAF08 enhances RSV-specific Th1 and CD8 T cell responses in newborn mice.** Neonatal CB6F1 mice were immunized with formulations as indicated at day of life (DOL) 4-7 and DOL 21-24. At DOL 35-38, spleen and lymph nodes were harvested, and cell suspensions prepared for flow cytometry staining with either a pre-F-specific MHC II tetramer, or a control tetramer containing the CLIP peptide, and stained for intracellular presence of Th1 transcription factor T-bet or Th2 transcription factor GATA3. **a** Representative histograms are shown. **b** Lymph node suspensions from all animals except saline-treated had a frequency of RSV tetramer-positive cells (closed circles) that was significantly higher than the frequency of control tetramer-positive cells (open circles,) Statistical comparisons to each corresponding control tetramer group employed two-tailed paired Student's $t$ test (*$p < 0.05$, **$p < 0.01$, ***$P < 0.001$).. No significant presence of RSV tetramer-specific T cells as compared to control tetramer-positive T cells was observed in the spleen. $n = 5$ newborn mice/group, shown as Box-whisker plots, center:mean, boxes:75th and 25th percentile, whiskers: minimum and maximum values. **c** Analysis of the relative percentage of T-bet+ Th1 cells and GATA3+ Th2 cells within the tetramer-positive gate revealed an increase in Th1 cells and a relatively lower abundance of Th2 cells in CAF08-immunized animals ($n = 5$ newborn mice/group) Statistical comparisons to SAL group employed two-tailed Mann–Whitney $U$ test (*$p < 0.05$, **$p < 0.01$, ***$P < 0.001$. **d** Draining lymph node cultures were stimulated with 1 μg/ml pre-F antigen for 24 h in the presence of Brefeldin A. Quantification of CD4+CD44+Cytokine+ (IFN-γ) cells confirmed a statistically significant generation of Th1 cells in CAF08-immunized mice. Detection of CD8+CD44+ Cytokine+ (IFN-γ and/or TNF-α) cells indicates induction of RSV-specific CD8+ T cells in CAF08-treated mice. Cytokine analysis indicates prominent production of Th2/anti-inflammatory cytokines (IL-5, −13, −10) in newborn mice as compared to adult mice. CAF08b reduced production of Th2/anti-inflammatory cytokines and increased that of IFN-γ, indicating enhancement of a Th1 response. Statistical comparisons to SAL group employed two-tailed Mann–Whitney $U$ test (*$p < 0.05$, **$p < 0.01$, ***$P < 0.001$ s ($n = 6$ newborn mice (CD4 analysis), 10 newborn mice (CD8 analysis and cytokine secretion analysis). Source data are provided as a Source Data file.

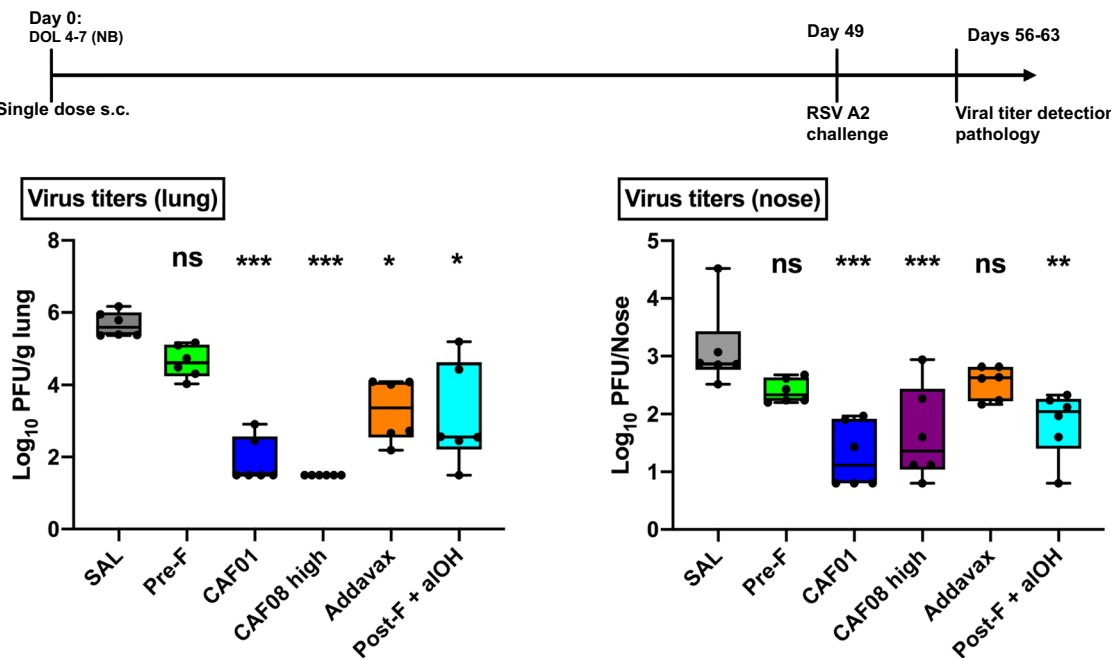

**Fig. 5 RSV pre-F protein combined with CAF08 protects mice from RSV infection.** Newborn CB6F1 mice were challenged with live RSV A2 strain after single immunization at birth with the formulations indicated. Viral load was measured in lungs and nose as described in the Methods. Adjuvantation of pre-F with CAF01 or CAF08 resulted in the greatest reduction in viral carriage in the lungs and nose. All statistical comparisons are Mann–Whitney $U$ test comparisons to saline groups ($n = 6$ mice/group, shown as Box-whisker plots, center:mean, boxes:75th and 25th percentile, whiskers: minimum and maximum values Statistical comparisons to SAL group employed two-tailed Mann–Whitney $U$ test (*$p < 0.05$, **$p < 0.01$, ***$P < 0.001$). Source data are provided as a Source Data file.

subsequently infected with live RSV A2 strain (Fig. 5). These data revealed that 4/6 mice in the CAF01 group and 6/6 mice in the CAF08 group did not have detectable levels of the virus in the lungs and 2-log reduced viral levels in the nose. Although a reduction in viral load was observed in the Addavax and Post-F + AlOH groups, CAF01- and CAF08- adjuvanted vaccines were significantly more effective. Infected mice had minimal weight loss, and histopathological evaluation of the lungs demonstrated an increase in inflammation, mucus, and eosinophil recruitment for all immunization groups compared to PBS control, with no significant difference between treatment groups- i.e., the vaccinated groups demonstrated inflammation indicating immune priming, but did not demonstrate any enhancement of disease, as assessed by weight monitoring, compared with RSV infection of unimmunized mice. (Supplementary Fig. 7).

Overall, the in vitro human phosphoproteomics platform and the murine immunization model presented here provide insight into age-specific features of PRR-mediated innate immune recognition and the adaptive immune bias towards a Th2-driven immune response also observed in humans in vitro and in vivo. Moreover, it reveals that combined stimulation of newborn DCs via TLR7/8 and Mincle results in protection against infection with RSV by induction of antigen-specific Th1 cells, CD8+T cells, and neutralizing antibodies.

## Discussion
The present work identifies the mechanism of action and in vivo applicability of adjuvant combinations that activate TLR7/8 and Mincle, which act in an age-specific manner to activate balanced Th1/Th2 polarizing responses by human neonatal antigen-

presenting cells[43]. We demonstrate that synergistic TLR7/8 and Mincle stimulation activates antigen cross-presenting pathways in vitro and protects against RSV infection in newborn mice, correlating with Th1-skewed CD4+ and CD8+ T cell responses towards the RSV pre-F antigen in vivo. The major challenge with the induction of early life immunity against RSV is the innate propensity of the immune system in early life to be skewed towards a Th2 phenotype, where a Th1 phenotype would be preferred for host defense against intracellular pathogens such as respiratory viruses. A Th2-mediated response may even worsen disease symptoms upon encounter of RSV, due to excessive infiltration of polymorphonuclear leukocytes (PMNs) into the lungs as well as Th2 cytokine-promotion of mucus production and airway hypersensitivity, both of which increase airway obstruction. Airway obstruction is thought to be the major factor causing severe illness and hypoxemia, especially in young infants with small airways.

Synergistic stimulation through the receptors TLR7/8 and Mincle enhances the production of Th1-favoring cytokines TNF-α, IL-18 and IL-1β in newborn MoDCs through restoration of NF-κB signaling and activation of the NLRP3 inflammasome, while also reducing the propensity of newborn MoDCs to secrete Th2/anti-inflammatory cytokines IL-10 and IL-12p40[43]. Given the age-specific nature of MoDC activation with R848 and TDB, we investigated whether additional signaling pathways were induced upon stimulation with R848 + TDB. We leveraged state-of-the-art label-free quantitative proteomics to determine changes in phosphorylation in newborn and adult MoDCs. Pathway overrepresentation analysis indicated age-specific synergy in cytokine signaling pathways such as TLR signaling, MAPK signaling and JAK-STAT pathways, as well as synergistic induction of endocytic pathways, suggesting a link between endocytosis and cytokine induction in newborn MoDCs. Indeed, synergistic TNF-α induction appeared dependent on endocytosis, with a proposed key role for PKC-δ. One possible explanation for these findings would be that cytokine signaling pathways are activated over a prolonged period as a result of prolonged delivery of R848 to endosomes. Future studies investigating the kinetics of signal transduction may shed further light on the role of the individual proteins identified.

In addition to establishing a link between endocytosis and cytokine production in newborn MoDCs, phosphoproteomic studies also suggested synergistic induction of antigen cross-presentation on MHC class I by R848 + TDB, both in newborn and adult cells. This is in accordance with the observation in our murine immunization study that RSV-specific CD8+ T cells were only induced in mice immunized with the liposomal adjuvant combining 3M-052 and TDB (CAF08).

Following confirmation of our mass-spectrometry observations using Western blotting and using target inhibition with small-molecule inhibitors and RNAi, a murine model was chosen to study the adaptive immune response induced by activation via TLR7/8 and Mincle, as this enabled the study of T cell responses against RSV induced by formulations containing either single adjuvants or combination adjuvants formulated with the RSV pre-F antigen. Analyses of pre-F-specific antibody, RSV neutralization, and B cell phenotype in the draining lymph node all illustrate that most RSV pre-F-containing formulations, including liposome formulations with single adjuvants, combination adjuvants, oil-in water emulsions or even soluble antigen alone can induce high Ab titers and germinal center B cell formation in newborn and adult mice. However, only 3M-052 (a TLR7/8 agonist)-containing formulations generated antibodies and germinal center B cells biased towards the Th1 induced IgG2a/c subset rather than IgG1. In contrast to IgG1, IgG2a/c can bind to FcγRI and FcγRIV and promote distinct effector functions[67,68].

TDB-containing formulations (CAF01, CAF08) elicited greater RSV-neutralizing ability. Altogether, the antibody titers and subclasses shown in this study imply an improved functionality of anti-RSV antibodies in mice immunized with CAF08. Improved antibody functionality is an important determinant of protection against RSV[69,70].

Although the vaccine formulations studied induced a similar quantity of RSV-specific CD4+ T cells, the phenotype of these cells differed markedly between the formulations. In this study we describe the development of an MHC class II tetramer for the detection of H2-IAb-restricted CD4+ T cell responses against an epitope in the F protein. This validated tetramer detects very robust frequencies of CD4+ T cells against a previously undescribed epitope. Recent mapping of T cell epitopes in the RSV proteome in C57BL/6 mice did not report this epitope[71], emphasizing the value of this tetramer for future murine immunization studies. Using this tetramer, we showed that immunization with CAF08 redirects the CD4+ T cell response to a Th1 phenotype in newborn mice, as opposed to the Th2 response that is typical for newborn mice and which was observed with single adjuvant-containing formulations. These observations are consistent with prior studies employing C-type lectin receptor agonists as adjuvants[45,46]. Other adjuvant combinations, such as (Alum + STING agonist) and IC31 (antimicrobial peptide + CpG-ODN), may also be able to elicit a Th1 response in newborn mice[72,73].

Our results suggest that the quality of the induced immune response is a critical parameter. Although all formulations studied, either with single adjuvants or with adjuvant combination (3M-052+TDB), induced similar levels of antibodies or CD4+ T cells, marked differences were noted in antibody isotype and T cell phenotype. The induction of RSV-specific Th1 cells as well as CD8+ T cells by CAF08 is relevant for protection against respiratory disease, as both Th2-driven pathology and lack of CD8+ T cells have been associated with enhanced disease after infection with RSV or after immunization with FI-RSV[74,75].

Our study features important strengths, including (a) the mechanistic investigation of the activity of an adjuvant combination towards primary human newborn leukocytes that can predict age-specific in vivo adjuvanticity; (b) validation of the relevance of the phosphoproteomic pathways identified via immunologic assays in leukocytes derived from different human study participants, and (c) confirmation of the mechanism of action of the adjuvant combination in murine studies in vivo.

Our study also has limitations, including: (a) As expected, human primary cells have more diverse signatures than uniform cell lines, resulting in donor-specific variability observed in PCA as well as volcano graphs; (b) the potential for false discovery, which was controlled for within the phosphoproteomic observations via extensive validation of individual phosphorylation events by western blotting, and confirmation of immune signatures via independent immunologic assays employing leukocytes from additional/distinct study participants in vitro and in mice in vivo; Our stringent mass-spectrometry analysis methodology included exclusion peptides with low phosphorylation localization probability, exclusion of biological replicates with a high degree of imputed data, and application of a permutation-based false-discovery rate to our statistical analysis. This has contributed to increased confidence in our findings. However, in our smaller validation cohort of adult samples, this has also led to inclusion of only two vehicle-treated replicates, reducing the confidence of the statistical analysis in this cohort. (c) that mice are no longer newborns when challenged post immunization; and (d) the murine RSV challenge model is semi-permissive, and carriage can occur without disease symptoms. Given their ability to mount different types of T-helper subsets relatively equally

without intrinsic bias, CB6F1 mice were employed to enable determination of the T-helper phenotype induced by our formulations. However, CB6F1 experience less weight loss after RSV infection than Th2-prone BALB/c[76]. Enhanced disease development after RSV infection in BALB/c mice is due in part to this intrinsic Th2 skew, but also in part due to a single immunodominant Vβ14+ CD4+ T cell response to an epitope in the G protein[77]. Because the primary objective of our study was to evaluate the ability of our formulation to overcome age-dependent Th2 skewing, a strain was chosen with more balanced abilities in development of Th1 and Th2 cells that would not be biased by a dominant response to a non-vaccine antigen. Although it is unclear if directing immune responses away from the neonatal Th2 bias could ameliorate lung pathology following infection, the robust ability of CAF08 to enhance Th1 responses and diminish viral carriage suggests that this adjuvantation system is well-suited for an RSV vaccine. Infiltration of eosinophils and other immune cells was observed in all immunization groups except the saline group, which did receive viral challenge. This is in accordance with prior literature[78], and likely contributes minimally to pathology, as evidenced by lack of weight loss. Given the Th2-polarizing tendency of Post-F + Alum and Pre-F + Addavax groups, increased lung pathology was anticipated, but was not observed. A possible explanation is that CB6F1 mice were used rather than BALB/c. Future studies to further assess safety and efficacy in early life would therefore benefit from utilizing a non-human primate model for RSV infection[79,80].

In conclusion, our study describes an adjuvanted vaccine formulation that overcomes the neonatal bias towards Th2-mediated immunity and provide robust Th1 and CD8-mediated protective immunity against RSV. Using state-of-the-art phosphoproteomics, this study also sheds light on the molecular basis of age-specific signal transduction upon receptor ligation and adjuvant synergy. The CAF08-adjuvanted pre-F vaccine induced an RSV-specific, Th1-polarized immune response in newborn animals Overall, this synergistic combination adjuvant has great potential for future development of precision adjuvanted pediatric vaccines to protect against RSV, influenza, coronaviruses, and other intracellular pathogens.

## Methods

**Agonists and inhibitors**. R848 (TLR7/8 agonist; InvivoGen) was reconstituted in sterile water at a concentration of 3 mM. TDB (Mincle agonist; Avanti Polar Lipids) was reconstituted in dimethyl sulfoxide (DMSO; Sigma-Aldrich) at a concentration of 20 mg/mL, heated at 60 °C for 30 s, then brought to 2 mg/mL using sterile Dulbecco's PBS (DPBS) without Ca2+, Mg2+, and phenol red (Life Technologies) and heated again at 60 °C for an additional 15 min. 3 M052 (TLR7/8 agonist) was a kind gift from Dr. Mark Tomai, 3M Drug Delivery Systems.

PKC-δinhibitor peptide (SFNSYELGSL; Anaspec Inc.), PKC inhibitor (Gö 6983; Abcam) and Clathrin inhibitor (Pitstop®1, and corresponding negative control; Abcam) was reconstituted in DMSO at 10 mM and used at 10 μM concentration in cell culture.

All agonists and inhibitors were verified to be endotoxin-free (<1 EU/ml), as measured by *Limulus amebocyte* lysate (LAL) assay per the manufacturer's instructions (Charles River). Sterile DPBS or DMSO were included in experiments at appropriate concentrations as a negative control when indicated.

**Liposome formulation**. Liposomes comprised of Dimethyldioctadecylammonium (DDA; Clauson-Kaas A/S) were prepared with or without adjuvants as indicated by using the thin film method combined with high shear mixing. Liposome formulations were characterized with respect to the average intensity-weighted hydrodynamic diameter (z-average), polydispersity index (PDI) and zeta-potential (Laser-doppler electrophoresis) essentially as previously described[81]. The final concentration of agonist-containing liposomes was 2.5/0.5 mg/mL DDA/TDB (DDA/TDB, or CAF01), 2.5/0.5 mg/mL (DDA/3M-052), 2.5/0.5/0.01 mg/mL (DDA/TDB;3M-052, or CAF08 low), and 2.5/0.5/0.05 (DDA/TDB/3M-052, or CAF08 high). RSV pre-F antigen (DS-CAV1) was incorporated into liposome formulations at a final concentration of 20 μg/mL by mixing with equal volumes of liposome formulation at room temperature for 1 h. Addavax, an MF-59-like oil-in-water emulsion was used as a control formulation, according to manufacturer's

instructions. The conformation of pre-F after incorporation into DDA liposome formulations was evaluated by binding to monoclonal antibodies AM-14, 5C4 and Motavizumab by dot blot (Supplementary Fig. 4). The binding of each monoclonal to pre-F after incorporation confirmed the intact presence of three major antigenic sites[15,16].

**Blood donors**. Non-identifiable cord blood samples were collected with approval from the Ethics Committee of the Beth Israel Deaconess Medical Center, Boston, MA (protocol number 2011P-000118) and The Brigham & Women's Hospital, Boston, MA (protocol number 2000-P-000117). All de-identified blood samples from adult (age 18–40 years old) study participants included in the experiments were collected with approval from the Ethics Committee of Boston Children's Hospital, Boston, MA (protocol number 307-05-0223), after written informed consent was provided. Blood samples were processed within 4 h (typically ~1–2 h), and anti-coagulated with 15 U/ml pyrogen-free heparin sodium (American Pharmaceutical Partners). The number of study participants used for each experimental approach is described in the text.

**Isolation of human mononuclear cells and moDC generation**. Heparinized blood from newborns and young adults (age 18-45 years) was centrifuged for 10 min at 500g, then the upper layer of clear yellow plasma was removed. This platelet-rich plasma was then centrifuged for 15 min at 3000g, and platelet-poor plasma was collected from the top and stored at +4˚C. The remaining blood was reconstituted to its original volume by resuspending in DPBS. Then, reconstituted blood was layered on to Ficoll-Hypaque gradients (Ficoll-Paque Premium; GE Healthcare) and centrifuged for 30 min at 500x g. After Ficoll separation, the mononuclear cell fraction was collected. Monocytes were then isolated from mononuclear cell fractions by positive selection with magnetic CD14 MicroBeads, performed according to the manufacturer's instructions (Miltenyi Biotec). Purity was routinely checked by flow cytometry and was always more than 98%. Isolated monocytes were seeded in 75 cm² tissue culture dishes for 5 d at 37˚C in a humidified incubator at 5% CO2 with 10⁶ cells/ml medium. Medium consisted of Cellgro DC medium (Cellgenix) supplemented with 1% penicillin-streptomycin (Life Technologies) and 10% autologous plasma. Cultures were supplemented with 50 ng/ml recombinant human IL-4 and 100 ng/ml recombinant human GM-CSF (R&D Systems). After 5 days, immature MoDCs were harvested by gently pipetting the loosely adherent fraction.

**Phosphoproteomics**. MoDCs were seeded in 6-well plates at $(3–5) \times 10^6$ cells per condition and rested for 30 min before stimulation with agonists as indicated for another 30 min. After stimulation, cells were harvested on ice and lysed in 50 mM Tris; 150 mM NaCl pH 7.7; 0.5% (v/v) Igepal CA-630 (Sigma-Aldrich); 1 μM DTT; 10% (v/v) glycerol; 1% (v/v) HALT protease inhibitor cocktail (Sigma-Aldrich); 1% (v/v) Phosphatase inhibitor cocktail 2 (Sigma-Aldrich); 1% (v/v) phosphatase inhibitor cocktail 3 (Sigma-Aldrich). Protein concentration in cell lysates was determined using BCA protein assay (Thermo Scientific). Samples were frozen at −80 °C and further processed batch-wise to reduce batch-effects. Equal protein amounts (1 mg protein) of all samples were denatured and digested with trypsin (Thermo Scientific) on Microcon-10 Centrifugal Filters (Millipore). Phosphorylated peptides were enriched by HPLC using a FeCl₃-charged ProPAC IMAC-10 column (Thermo Scientific) as described previously, resulting in a typical enrichment of phosphopeptides from <1% in the original lysate to >40% in the enriched fraction[82]. Phosphopeptide-enriched fractions were desalted using a 1 cc Oasis HLB cartridge (Waters Corp.). Samples were analyzed on a nanoflow ultrahigh-performance liquid chromatography (UPLC) system (400 Series, Eksigent/Sciex) hyphenated with a quadrupole-Orbitrap mass spectrometer (Q Exactive; Thermo Scientific). The Q Exactive mass spectrometer was run in positive-ion mode. Full scans were carried out at a resolution of 70 K with an automatic gain control (AGC) target of $3 \times 10^6$ ions and a maximum injection time of 120 ms, using a scan range of 350 to 2,000 m/z. For tandem MS (MS/MS) data acquisition, a normalized collision energy value of 27 was used. Scans were carried out at a resolution of 35 K with an AGC target of $3 \times 10^6$ ions and a maximum injection time of 120 ms. The isolation window was set to 2 *m/z*. An underfill ratio of 0.5% was set and a dynamic exclusion value of 20 s applied. RAW files generated with XCalibur software (version 2.2; Thermo Scientific), were analyzed using Maxquant (version 1.5.6.5) for identification and quantification of phosphopeptides. Phosphorylation (STY) and oxidation (M) were used as variable modifications and carbamidomethylation as a fixed modification. Match between run and label-free quantification (LFQ) were enabled. iBAQ was calculated for all samples[83]. Only peptides identified as phosphorylated were included in subsequent analysis. The mass spectrometry proteomics data have been deposited to the ProteomeXchange Consortium via the PRIDE[84] partner repository with the dataset identifier PXD025568. Maxquant peptide identification and quantification was done using the Uniprot human genome reference database (UP000005640).

As shown in Supplementary Fig. 1, the 'phospho (STY) Sites.txt' file generated by Maxquant, from either the newborn ($n = 9$) dataset or from the adult ($n = 4$) dataset, was used for further analysis of treatment-induced changes in phophopeptide abundance, using Perseus software (version 1.6.15.0[85]). The peptide datasets were first curated by removal of reverse hits, potential contaminants,

peptides with a phosphorylation site probability < 0.75, and peptides that were not identified in ≥ 50% of the samples (5/9 donors for the newborn dataset, 2/4 donors for the adult dataset) in at least one of the treatment groups (Supplementary Fig. 1A). Samples in which the remaining peptide intensities consisted of <25% of valid values were excluded from subsequent analysis. Intensities were log2-transformed and missing values were imputed from normal distribution. Unpaired two-tailed Student's *t* test was used in combination with a permutation-based fold-change threshold cutoff (FDR = 0.05, σ=0.2) to determine which phosphopeptides were differentially induced by treatment as compared to vehicle controls (Supplementary Fig. 1B, Supplementary Datasets 1–6). Because most proteins can be phosphorylated on multiple sites, and different phosphorylation sites within a protein can exert different effector functions, each phosphorylation site was considered an independent event. Pathway overrepresentation analysis (ORA) was assessed using *InnateDB*[86], weighing peptide fold-change as well as p-value. For ORA, multiple protein function databases were mined; KEGG[47], REACTOME[48], PID NCI[49], BioCarta[50], NetPath[51] and INOH[52].

**Western blotting**. Cell lysates generated from treated moDCs as above were precipitated by addition of 4x volume of acetone (Sigma-Aldrich) and incubation at −20 °C for 24 h. Samples were centrifuged for 15 min at 12,000*g* at 4 °C. Pellets were resuspended in LDS sample buffer (Invitrogen) at a concentration of 2.5 mg/mL. 25 μg of each lysate was analyzed by SDS-PAGE on 10% Bis-Tris gel (Invitrogen) or 12.5% Bis-Tris gel containing 50 μmol/L SuperSep Phos-tag™ reagent (Wako Chemicals) Separated proteins were transferred to PVDF membrane (Invitrogen) and incubated with primary antibodies as indicated; anti-Sec22b (Novus Biologicals), anti-PKC delta (Abcam), anti-PKC delta phospho-S645 (Abcam), anti-STAT1 (Abcam), anti-STAT1 phospho-S727 (Abcam), anti-GAPDH (Abcam), anti-STX4 (BD Biosciences), anti-PSME2 (Invitrogen), anti-NCF2 (Invitrogen) for 18 h at 4 °C. Membranes were washed and incubated with HRP-conjugated anti-rabbit IgG (Invitrogen) or anti-mouse IgG (Cell Signaling Technology) at room temperature for 1 h. Chemiluminescent substrate (Super-Signal™ West Pico PLUS; Thermo Fisher) was used for photodetection of target proteins.

**Cytokine measurements**. Culture supernatants from human moDCs or from murine splenocyte or lymph node cultures were stored at −20 °C until analysis. Human TNF-α was quantified using the Opteia ELISA set (BD Biosciences). ELISA plates were read on a Versamax microplate reader with SoftMax Pro Version 5 (both from Molecular Devices). Murine TNF-α, IFN-γ, IL-17, IL-5, IL-13 and IL-10 were measured using a multiplexing bead array (Millipore). Sample fluorescence was acquired on a Flexmap 3D analyzer with xPONENT version 4.2 software (Luminex). Results were converted into pg/ml values by 4-point log curve fitting on kit-provided standards using Milliplex Analyst version 3.5.5.0 software.

**Immunization**. All experiments involving animals were approved by the Animal Care and Use Committee of Boston Children's Hospital and Harvard Medical School (protocol numbers 15-11-3011 and 16-02-3130) and studies at Statens Serum Institut were approved by the Danish governmental Animal Experiments Inspectorate (protocol number 2014-15-2934-01065). CB6F1 mice were generated by cross-breeding female BALB/c mice with male C57BL/6 mice (Charles River). First generation offspring (CB6F1) were immunized with indicated formulations. Adult cohorts (6-12 weeks of life) were immunized s.c with 100 μL formulation and again 21 days later. Agonist and pre-F concentrations are indicated in figure legends. All adult formulations contained 2 μg pre-F antigen. Blood was drawn for antibody titer determinations 14 days after each immunization. Newborn cohorts were immunized in the first week of life with half the adult dose (50 μL) s.c formulation and again 21 days later. Blood was also drawn for antibody titer determinations 14 days after each immunization.

**RSV challenge**. For the challenge study, CB6F1 mice were immunized in the first week of life with a single dose of each formulation as indicated, and subsequently challenged intranasally with 5×10^6 PFU of wild-type RSV A2 in a volume of 100 μL under isoflurane (2-3%) anesthesia at 8 weeks of age. These studies, conducted at the Vaccine Research Center (VRC) at NIH, were approved by the VRC Animal Care and Use Committee (protocol VRC-17-716). Several animals were euthanized on Days 5 and 8 post-challenge to assess viral titer in the nose and lungs (Day 5), and lung histopathology (Day 8). The remaining 5 animals per group were monitored for weight loss through Day 14. To measure viral load, noses and lungs were homogenized (using chilled mortar and pestle with ground glass for noses, and gentleMACS dissociator for lungs), and clarified homogenates were titrated by serial dilution on HEp-2 cells as previously described (PMID 22144888). Virus titers are reported as the log_10 TCID50/g of tissue or per nose.

**Lung Histopathology and Immunohistochemistry**. Lung samples from mice were processed per a standard protocol. Briefly, the tissues were fixed in 10% neutral buffered formalin, processed with Leica ASP6025 tissue processor (Leica Microsystems), embedded in paraffin, sectioned at 5 μm, and blocked in paraffin for histological analysis. Tissue sections were stained with hematoxylin and eosin (H&E) or PAS (Periodic Acid-Schiff) histochemical stains for routine

histopathology and the for the location of Goblet cells and presence of airway mucin accumulation, respectively. Sections were examined, in a blinded manner, by a boarded-certified veterinary pathologist using an Olympus BX51 light microscope and photomicrographs were taken using an Olympus DP73 camera.

For immunohistochemical (IHC) evaluation, formalin-fixed paraffin-embedded tissues sections (5 μm) obtained from mice, were used to perform immunohistochemical staining using an antibody which identifies eosinophils in tissue; A Goat Polyclonal EMBP (Santa Cruz; sc-33938) antibody at a dilution of 1:500. Staining employed the Bond RX (Leica Biosystems) platform according to manufacturer-supplied protocols. Briefly, 5μm-thick sections were deparaffinized and rehydrated. Heat-induced epitope retrieval (HIER) was performed using Epitope Retrieval Solution 1, pH 6.0, heated to 100 °C for 20 min. The specimen was then incubated with hydrogen peroxide to quench endogenous peroxidase activity prior to applying the primary antibody. Detection with DAB chromogen was completed using the Bond Polymer Refine Detection kit (Leica Biosystems CAT# DS9800). Slides were finally cleared through gradient alcohol and xylene washes prior to mounting and placing cover slips.

**Anti-pre-F antibody titer determination**. Microtiter plates were coated with 1 μg/mL pre-F. Serum from immunized mice was incubated in 10-fold serial dilutions onto coated plates at room temperature for 2 h. Pre-F-specific antibody isotypes were detected using HRP-labeled anti-mouse IgG (AH Diagnostics), IgG1 (Southern Biotech), IgG2a (AH Diagnostics), or IgG2c (Southern Biotech). End-point titers were subsequently derived from dilution curves using adaptation of Frey et al.[87]. Briefly, titer was defined as the reciprocal of the highest serum dilution generating a reading above a cutoff defined as Cutoff = $\mu X$ + SD $\tau \sqrt{(1 + (1/n))}$ where $\mu X$ is the mean of independent control sera readings, SD is the standard deviation, n is the number of independent controls, t is the $(1-a)$th percentile of the one-tailed t-distribution with $v = n-1$ degrees of freedom. As an adaptation, all individual curves were fit to 4-point log curve using nonlinear regression analysis. The approximate serum dilution (Y) that intersected with the above-calculated cutoff was then calculated using: Cutoff=Bottom + (Span)/(1+10^((LogIC50-Y)*Slope)).

**High-throughput fluorescence plate reader neutralization assay**. A total of 2.4 × 10^4 HEp-2 cells/well in 30 μL culture medium were seeded in 384-well black optical-bottom plates (Nunc®384-well plates, Thermo Scientific). Antibodies were diluted four-fold, starting at 100 μg/mL. An equal volume of recombinant mKate-RSV A2 or mKate-RSV B 18537 was then added and incubated at 37 °C for 1 h. After incubation, 50 μL of the antibody–virus mixture was added to the HEp-2 cells and incubated at 37 °C for 22–24 h. After incubation, the fluorescence intensity of each well was measured using a microplate reader at an excitation of 588 nm and an emission of 635 nm (*SpectraMax Paradigm*, Molecular Devices). Neutralization IC50s were calculated using GraphPad Prism (GraphPad Software Inc.).

**Development of an RSV pre-F MHC II tetramer**. The amino acid sequence of the RSV pre-F antigen[16] was used to predict epitopes binding to any of the MHC class II alleles of CB6F1 mice. Sequences were considered putative epitopes if predicted to bind to either H2-IAd, H2-IEd or H2-IAb in the top 2% rank percentile using NetMHCIIpan 3.2 server[88] and a binding score of ≥10 using SYFPEITHI[89]. Peptide sequences with a compatible MHC facing agretope were subsequently evaluated in silico for putative TCR binding strength[90]. Further elimination of sequences containing cysteine residues, likely to cause tetramer unfolding, resulted in the selection of peptide sequences tested in vitro to induce activation of CD4+ T cells in splenocyte cultures from immunized mice. An RSV pre-F peptide, consisting of residues 235-249 (REFSVNAGVTTPVST), was synthesized and purified to >98% purity by Genscript (Piscataway, NJ, USA). A PE-labeled recombinant H2-IAb tetramer bound to either synthesized RSV F235-249 sequence, or the human Class II-associated invariant chain peptide (CLIP) was produced by the NIH Tetramer Core Facility at Emory University. The tetramer specificity was validated on lymph node cell suspensions from immunized CB6F1 mice. Staining of CD4+ cells from mice immunized with pre-F -containing formulations consistently stained positive (1-2%) with the RSV F235-249 tetramer, but not the CLIP tetramer. Cells from saline-injected mice did not stain positive with the RSV F235-249 tetramer. In addition, as further evidence of tetramer specificity, single cell sorting of tetramer-positive cells and clonal expansion with anti-CD3/CD28-coated microbeads (Thermo Fisher) generated a fully tetramer-positive CD4+ T cell clone (Supplementary Fig. 4), further evidence of tetramer specificity.

**Ex vivo recall assay**. Cell suspensions from murine brachial lymph nodes and spleens were generated by mashing through a 70 μM strainer (Thermo Fisher). In spleen suspensions, erythrocytes were lysed by 2 min of incubation in ammonium chloride-based lysis buffer (BD Biosciences). Cells were then counted and plated (1–2) × 10^6 per well in a flat-bottom 96-well microtiter plate. Cells were incubated with or without recombinant RSV pre-F antigen (1 μg/mL) in 200 μL RPMI 1640 + 10% (v/v) heat-inactivated fetal calf serum (Hyclone), 5 × 10^−6 M β-mercaptoethanol, 1% (v/v) penicillin-streptomycin, 1% (v/v) sodium pyruvate, 1 mM L-glutamine, 100x dilution of non-essential amino acids and 10 mM HEPES (all from Life Sciences). Cells were either incubated for 72 h for measurement of secreted

cytokines, or for 24 h in the presence of 0.1% (v/v) BD GolgiPlug (BD Biosciences) for flow cytometry.

**Flow cytometry**. For phenotypic analysis of germinal center B cells and T follicular helper cells in spleen and lymph node suspensions, cells were stained for 30 min at 4 °C with anti-GL7 (FITC), anti-IgG1 (PE), anti-B220 (PerCP-Cy5.5), anti-CD38 (PE-Cy7), anti-IgG2c (APC) and anti-IgD (BV786), or with anti-CD4 (FITC), anti-CD279 (PE), anti-B220 (PerCP-Cy5.5), and anti-CXCR5 (BV421), all from BD Biosciences.

For phenotypic analysis of antigen-specific CD4+ T cells, cell suspensions were incubated for 1 h at 37 °C with RSV F235-249 tetramer or CLIP tetramer as control (PE). Samples were subsequently fixated in 1% paraformaldehyde (Alfa Aesar) for 30 min at 4 °C. Samples were the permeabilized using BD Perm/Wash (BD Biosciences) and stained with anti-T-bet (Alexa 647) and anti-GATA3 (Alexa 488) both from BD Biosciences).

For evaluation of ex vivo recall cultures, cells were fixated in 1% paraformaldehyde for 30 min at 4 °C, permeabilized using BD Perm/Wash and stained with anti-CD4 (V450 or BV786), anti-IL-2 (APC-Cy7), both from BD Biosciences, and anti-CD44 (FITC), anti-CD8 (PerCP-Cy5.5), anti-IFN-γ (PE-Cy7), anti-TNF-α (PE), anti-IL-17 (APC) (eBioscience).

**Statistical analysis**. Data were analyzed and graphed using Prism for MacIntosh v. 7.0 (GraphPad Sofware). Tests used for statistical comparisons are indicated in figure legends. A p-value <0.05 was considered significant. $^*p < 0.05$, $^{**}P < 0.01$, $^{***}p < 0.001$.

**Reporting summary**. Further information on research design is available in the Nature Research Reporting Summary linked to this article.

## Data availability
The mass spectrometry proteomics data have been deposited to the ProteomeXchange Consortium via the PRIDE[84] partner repository, PXD025568 is the dataset identifier. Source data are provided with this paper.

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

## Acknowledgements

The authors would like to thank our study participants for their contribution to this study. We are also very grateful for the labor and delivery staff at the Brigham and Women's Hospital and Beth Israel Deaconess Medical Center for their assistance with cord blood donations. We gratefully acknowledge Dr. Camilia Martin for expert assistance in coordinating cord blood collection protocols. This study was funded by US National Institutes of Health (NIH)/National Institutes of Allergy and Infectious Diseases (NIAID) grant on Molecular Mechanisms of Combination Adjuvants U01AI124284-01 (OL). O.L., H.S., and S.D.v.H. are also supported by the Department of Health and Human Services, Adjuvant Discovery Program contract no. HHSN272201400052C; NIAID grant on Human Immunology Project Consortium U19AI118608 and an internal award from the Boston Children's Hospital Department of Pediatrics to the *Precision Vaccines Program*. S.D.v.H is also supported by the Bill and Melinda Gates Foundation, grant INV-004886 and previously by NIH/NICHD grant, 5T32HD055148-10.

## Author contributions

S.D.v.H., G.K.P., B.S.G., H.S., D.C., P.A., and O.L. designed the study. S.D.v.H, G.K.P., A.K., T.J.R., S.M., I.N.M., M.M., M.L., J.P., F.B., S.D.G., E.M.S.B., S.A., and M.H. conducted the experiments. S.D.v.H. wrote the manuscript, all authors edited the manuscript.

## Competing interests

S.D.v.H., F.B. and O.L. are named inventors on patents describing early life vaccine adjuvant compositions. B.S.G. is named an inventor describing RSV prefusion F as a candidate vaccine. F.B. has signed consulting agreements with Merck Sharp & Dohme Corp. (a subsidiary of Merck & Co., Inc.), Sana Biotechnology, Inc., and F. Hoffmann-La Roche Ltd. These commercial relationships are unrelated to the current study. The rest of the authors declare no competing interests.
