## [Peer Review File · Nature Communications]

CAF08 adjuvant enables single dose protection against Respiratory Syncytial Virus infection in murine newbornsREVIEWER COMMENTS

Reviewer #1 (Remarks to the Author):

The search for a vaccine against RSV has been ongoing for many years with little success and thus data such as these are important and timely, especially with the boost vaccine research (especially in respiratory viruses) has received recently due to the COVID-19 pandemic.

This is an interesting and well written article. It investigates an RSV vaccine with various adjuvants using both in vitro techniques on human blood and an in vivo murine model. The investigators show that the vaccine/adjuvant (CAF-08) are immunogenic and has a TH1 biased T cell response. This is important as in human infants a TH2 skewed response is associated with more severe RSV disease. In the mouse model the vaccine showed a reduction in RSV viral load in the nose and lungs compared with control (saline).

I have the following comments:

- As is highlighted in the discussion, the genetic background of the mice has significant implications on the findings in the study (as with all murine models of RSV). In the results section even the mice who received “placebo” (rather than the vaccine) did not lose weight even though they had high levels of RSV in their nose and lungs. This implies they are less permissive to RSV than other mouse strains (e.g. BALB/c). How much does this impact on the lung pathology results & potentially the “immunology” results? i.e. would the findings be different if BALB/c mice were used?
- I am not clear about the comment that there was “an increase in inflammation, mucus, and eosinophils in the immunised group” (lines 270-4; 364-367). If I am interpreting the results correctly, mice who were immunised (with any of the vaccines) had reduced viral loads but more lung inflammation than the mice who received placebo who had high levels of virus but no inflammation. This seems contrary to the sentence that “the vaccinated groups did not demonstrate any enhancement of disease compared with RSV infection of unimmunized mice”. Can you explain this?
- There was no quantitative assessment of lung inflammation. Could a more detailed assessment be made of this other than the sentence in the text stating “more inflammation etc”? This is especially important as enhanced disease is a particular worry with RSV vaccines.
- One of the limitations noted is that the newborn mice are no longer “newborn” when challenged with RSV. How might this impact the findings, in particular the TH1-TH2 skew?
- Line 78- add consider changing “adults or elderly” to “young or elderly adults”
- L202- “adult doses of 100ml”- should be 100uL?
- L364-365 Please rephrase the sentence as it doesn’t really make sense.
- L434-5- Ensure correct formatting (sub/superscript) of CO₂, cm² etc.
- Please be consistent with abbreviations for minutes, hours, days throughout the manuscript e.g. minutes are written as “m”, “min” or “minutes” in different places.
- L461- need a space in “to²”.
- L546- This sentence is a repeat of the sentence at L532-4.
- L703- Write “LN” in full.
- Figure 3- why is the SAL group not included in the IgG2a/IgG1 graphs? It is possible to calculate the ratio for this group.
- Supp Fig 2. Legend- “incorporation” should be “incorporation”.
- Supp Fig 4. Legend- There is no note in the legend of either (D) or (E).
- It is unusual to include reference to Figures in the discussion section.
- Overall, the references seem appropriate. Number 75 in the reference list needs to have the title added and be edited correctly. Check all the other refs are edited correctly.

Reviewer #2 (Remarks to the Author):

In this manuscript the authors describe a cationic adjuvant formulation (CAF)-08, a liposomal vaccine formulation tailored to induce Th1 immunity via synergistic engagement of Toll-like Receptor 7/8 and the C-type lectin receptor Mincle, and performed quantitative label-free phosphoproteomics of human dendritic cells after combined stimulation through TLR7/8 and Mincle.

I feel that I am best qualified to review the phosphoproteomic data and focus my review on that. There are major points regarding phosphoproteome analysis that need to be addressed:

-how much starting material used for phosphopeptide enrichment and how pure phosphopeptide populations were obtained?

-what was the database used in MaxQuant search? Why was LFQ activated when focus is on phosphopeptides? What file from the Combined-folder was used for Perseus analysis, and how was Perseus analysis done? Now it is only mentioned that 'Log2- normalized fold changes over vehicle-treated condition and corresponding p-value (Paired Student's t-test) for each phosphopeptide were derived using Perseus'. How were missing values taken into account? How was the peptide-level data normalized? Was there any filtering applied regarding the phosphopeptide site localization? How well did the sample groups separate from each other?

-no data on the MaxQuant and Perseus analysis is shown (tables), and the data is not deposited into proteomics data repository like ProteomeXchange, this makes it impossible to evaluate the quality of the data

-in results the authors state that 'Between 250 and 900 phosphorylation events were identified and quantified per sample. In a sample size of nine newborn donors and five adult donors, the quantity of 186 to 467 phosphorylation sites changed their phosphorylation status in a statistically significant manner, depending on the treatment and age group' and refer to suppl fig1. Again, there is no data on the reproducibility of the results, and in my opinion the numbers of phosphopeptides identified and quantified are not very impressive posing questions on the quality of the data. With optimized phosphopeptide enrichment strategy combined with high-resolution LC-MS/MS it should be possible to quite easily identify thousands of phosphopeptides from total cell lysates.

-In Suppl fig 1 the volcano plots for adult cells (Suppl fig 1B) seem highly biased, why is that?

-Fig 1C-D show phosphoproteomic results for two phosphopeptides, but again it is not clear what data from MaxQuant search has been used for this. Fig 1E shows 'confirmation of MS results with western blotting', however, the WB is not of very good quality and no fold-change values have been calculated to compare that with the MS-data

-Fig 1F-G (Inhibition of endocytosis with a clathrin inhibitor): for these experiments Clathrin inhibitors Pitstop®1, 2 were used. How specific are they? Pitstop2 has been shown to be a potent inhibitor of also clathrin-independent endocytosis

-Fig 2A-C can only be evaluated after having more details on the phosphoproteome analysis, and Fig 2B WB images are missing sample group info

Experiments with liposomal formulations, CD8+ T cell activation studies in vitro and in vivo newborn immunization model seem to be well done, but how well they are linked to the phosphoproteome data should be re-evaluated after having more details on the phosphoproteomics. Overall, these two parts of the manuscript are somewhat distant to each other in the current version of the manuscript.

We greatly appreciate the expert feedback provided to us by the reviewers and we have fully addressed their feedback with additional experiments and changes to the manuscript. As outlined below, we have responded to each of the reviewer comments, and have indicated for each point where and how the manuscript was changed.

RESPONSE TO REVIEWER COMMENTS:

Reviewer #1 (Remarks to the Author):

The search for a vaccine against RSV has been ongoing for many years with little success and thus data such as these are important and timely, especially with the boost vaccine research (especially in respiratory viruses) has received recently due to the COVID-19 pandemic.

This is an interesting and well written article. It investigates an RSV vaccine with various adjuvants using both in vitro techniques on human blood and an in vivo murine model. The investigators show that the vaccine/adjuvant (CAF-08) are immunogenic and has a TH1 biased T cell response. This is important as in human infants a TH2 skewed response is associated with more severe RSV disease. In the mouse model the vaccine showed a reduction in RSV viral load in the nose and lungs compared with control (saline).

Reply from the authors: Thank you for this positive assessment and for your expert feedback. Our response to each of your comments can be found below.

I have the following comments:

- As is highlighted in the discussion, the genetic background of the mice has significant implications on the findings in the study (as with all murine models of RSV). In the results section even the mice who received “placebo” (rather than the vaccine) did not lose weight even though they had high levels of RSV in their nose and lungs. This implies they are less permissive to RSV than other mouse strains (e.g. BALB/c). How much does this impact on the lung pathology results & potentially the “immunology” results? i.e. would the findings be different if BALB/c mice were used?

Reply from the authors:

Thank you giving us the opportunity to further expand on this important topic. BALB/c mice are indeed slightly more permissive to RSV than CB6F1 mice. While one of the prime underlying causes of enhanced disease development after RSV infection in BALB/c mice is the intrinsic skew towards Th2 development that such mice have, it is also heavily influenced by a single immunodominant V β 14+ CD4+ T cell response to an epitope in the G protein. The primary objective of our study was to evaluate the ability of our formulation to overcome age-dependent Th2 skewing, so we chose a strain with more balanced abilities in development of Th1 and Th2 cells that would not be biased by a dominant response to a non-vaccine antigen. CB6F1 mice are able to respond to a broader range of T cell epitopes and have an intermediate, more balanced immune response compared to each of the parental strains. We do acknowledge that performing our studies in a single strain is a limitation of our study, and that using different strains of mice, or different challenge doses could alter the immune response and subsequent disease, but we do not anticipate that it would affect the relative differences between immunization groups.

To address this important issue, we have edited the discussion section on lines 401-406 to reflect this information.

I am not clear about the comment that there was “an increase in inflammation, mucus, and eosinophils in the immunised group” (lines 270-4; 364-367). If I am interpreting the results correctly, mice who were immunised (with any of the vaccines) had reduced viral loads but more lung inflammation than the mice who received placebo who had high levels of virus but no inflammation. This seems contrary to the sentence that “the vaccinated groups did not demonstrate any enhancement of disease compared with RSV infection of unimmunized mice”. Can you explain this?

Reply from the authors:

We apologize if these sections were written in a way that appears contradictory. We indeed observed infiltration of immune cells in the lungs in immunized groups, although it is difficult to conclude whether this would lead to enhancement of disease, as none of the animals were observed to have any weight loss or other signs of disease. We have re-written the statement (lines 305-308) as follows: “the vaccinated groups demonstrated inflammation indicating immune priming, but did not demonstrate any enhancement of disease, as assessed by weight monitoring, compared with RSV infection of unimmunized mice.”

- There was no quantitative assessment of lung inflammation. Could a more detailed assessment be made of this other than the sentence in the text stating “more inflammation etc”? This is especially important as enhanced disease is a particular worry with RSV vaccines.

Reply from the authors:

Quantitative assessment of lung inflammation has now been provided to the manuscript as well, see Supplementary Figure 5C. Thank you for this suggestion. Given the limitations of the semi-permissive mouse model, it is our opinion that the biggest strength of the challenge study was the ability to evaluate reduction in viral carriage.

- One of the limitations noted is that the newborn mice are no longer “newborn” when challenged with RSV. How might this impact the findings, in particular the TH1-TH2 skew?

Reply from the authors:

This is indeed an important point; the brevity of the neonatal period in mice does not allow for post-immunization challenge to occur prior to adulthood. However, a major strength of our study is that we were able to demonstrate a preferential antigen specific Th1 induction upon newborn immunization via the use of our novel developed MHC tetramer. Analysis of tetramer-negative T cells in the spleen and lymph node clearly shows that at the time of RSV-specific Th1 induction, the rest of the CD4 compartment remained heavily Th2 skewed.

For the challenge study, the T cells were instructed by a single immunization at birth, resulting in antigen-specific Th1 cells in CAF08-immunized mice. In the other groups, naturally developing Th1 potential with age may indeed have contributed to lack of disease symptoms upon challenge.

- Line 78- add consider changing “adults or elderly” to “young or elderly adults”

Reply from the authors:

Thank you for this suggestion. We have changed the term ‘adults’, and ‘elderly’ ‘young adults’ and ‘elderly adults’, respectively, in line 80, as well as line 132

- L202- “adult doses of 100ml”- should be 100uL?

Reply from the authors:

This should indeed be 100uL. We have corrected this.

- L364-365 Please rephrase the sentence as it doesn’t really make sense.

Reply from the authors:

We have rephrased this sentence as follows (now lines 412-414)

“Given the Th2-polarizing tendency of Post-F+Alum and Pre-F+Addavax groups, increased lung pathology was anticipated, but was not observed. A possible explanation is that CB6F1 mice were used rather than BALB/c”.

- L434-5- Ensure correct formatting (sub/superscript) of CO₂, cm² etc.

Reply from the authors:

We have corrected these.

- Please be consistent with abbreviations for minutes, hours, days throughout the manuscript e.g. minutes are written as “m”, “min” or “minutes” in different places.

Reply from the authors:

We have checked manuscript for consistency and adjusted all to 'minutes' and 'hours'.

- L461- need a space in "to2".

Reply from the authors:

We have added a space between 'to' and '2'

- L546- This sentence is a repeat of the sentence at L532-4.

Reply from the authors:

Our apologies, we have removed the second sentence.

- L703- Write "LN" in full.

Reply from the authors:

We have written 'lymph node' instead of LN.

- Figure 3- why is the SAL group not included in the IgG2a/IgG1 graphs? It is possible to calculate the ratio for this group.

Reply from the authors:

We have added this ratio for the SAL group in our revised manuscript. We have also added a sentence to the results section to indicate that this ratio should not be over-interpreted in the SAL group, as both the IG2a and the IgG1 titers were in the lower regions of detection (Lines 237-240).

- Supp Fig 2. Legend- "icorporation" should be "incorporation".

Reply from the authors:

We have made this correction in the figure legend. In the revised manuscript, this is no Supplementary Figure 4.

- Supp Fig 4. Legend- There is no note in the legend of either (D) or (E).

Reply from the authors:

We have added a description of Panel D and E in the legend of this figure, which is now Supplementary Figure 6.

- It is unusual to include reference to Figures in the discussion section.

Reply from the authors:

We have rewritten the Discussion section to discuss the implications of our observations without references to Figures.

- Overall, the references seem appropriate. Number 75 in the reference list needs to have the title added and be edited correctly. Check all the other refs are edited correctly.

Reply from the authors:

We have corrected this and checked all other references. Thank you for your thorough review of our manuscript.

Reviewer #2 (Remarks to the Author):

In this manuscript the authors describe a cationic adjuvant formulation (CAF)-08, a liposomal vaccine formulation tailored to induce Th1 immunity via synergistic engagement of Toll-like Receptor 7/8 and the C-type lectin receptor Mincle, and performed quantitative label-free phosphoproteomics of human dendritic cells after combined stimulation through TLR7/8 and Mincle.

I feel that I am best qualified to review the phosphoproteomic data and focus my review on that. There are major points regarding phosphoproteome analysis that need to be addressed:

Reply from the authors:

Thank you for your detailed review of the proteomic data. Replies to each of your points can be found below.

-how much starting material used for phosphopeptide enrichment and how pure phosphopeptide populations were obtained?

Reply from the authors:

One milligram protein starting material was used for phospho-peptide enrichment.

After Maxquant identification and quantification of peptides, more than 40% of total peptide intensity consisted of phospho-peptides, a considerable enrichment from the starting material. Of note, only peptides identified to be phosphorylated by Maxquant were included in subsequent analysis.

We have included this information in the methods section of the manuscript (Lines 505, 508-509, 524-525), and have included additional supplementary excel tables of all identified phosphopeptides and their log2 intensities across all samples in the study.

-what was the database used in MaxQuant search? Why was LFQ activated when focus is on phosphopeptides? What file from the Combined-folder was used for Perseus analysis, and how was Perseus analysis done? Now it is only mentioned that 'Log2- normalized fold changes over vehicle-treated condition and corresponding p-value (Paired Student's t-test) for each phosphopeptide were derived using Perseus'. How were missing values taken into account? How was the peptide-level data normalized? Was there any filtering applied regarding the phosphopeptide site localization? How well did the sample groups separate from each other?

Reply from the authors:

We thank the reviewer for emphasizing the importance of detailed reporting of our proteomics analysis pipeline. We have updated the methods section of the manuscript to include the following information (Lines 528-538):

“Maxquant peptide identification and quantification was done using the Uniprot human genome reference database (UP000005640). The ‘evidence’ file generated by Maxquant was used to evaluate peptide sequence, protein designation, modifications, and intensity. When a peptide was identified in two or more samples, but not all, its intensity in samples where it was not identified was imputed to half the lowest detected intensity of any peptide in that sample. Intensities of overlapping sequences containing the same phosphorylation were added together. Because of multiple sites, and different phosphorylation sites within a protein can exert different effector functions, each phosphorylation site was considered an independent event. In Perseus, peptide intensities were expressed as fold-change over vehicle-treated samples, log2-transformed and z-score normalized within each sample to control for inter-sample variation.”

Because peptide intensity is used to calculate LFQ, this indeed did not need to be activated. This was done out of habit, but it does not affect the subsequent analysis of phosphopeptides.

In addition, we have included a PCA analysis as Supplementary Figure 1C to illustrate the separation of the samples in a two-dimensional field. PCA analysis revealed that samples cluster predominantly by age group, with minimal separation by treatment. This actually emphasizes the relevance of our pathway analysis approach, because it reveals significantly induced signaling pathways by treatment, even in the presence of large volumes of age-specific and donor-specific signaling events.

-no data on the MaxQuant and Perseus analysis is shown (tables), and the data is not deposited into proteomics data repository like ProteomeXchange, this makes it impossible to evaluate the quality of the data.

Reply from the authors:

Thank you for these suggestions. The mass spectrometry proteomics raw data, as well as ‘evidence’ output files from Maxquant have been deposited to the ProteomeXchange Consortium repository via the PRIDE partner repository with the dataset identifier PXD025568. This information has also been added to the manuscript (Line 525-527).

You may access the raw dataset prior to publication using the following credentials:

Reviewer account details:

Username: reviewer_pxd025568@ebi.ac.uk

Password: YbAllnYn

In addition, as mentioned above, we have included supplementary tables with all identified phosphopeptides and their intensities.

-in results the authors state that ‘Between 250 and 900 phosphorylation events were identified and quantified per sample. In a sample size of nine newborn donors and five adult donors, the quantity of 186 to 467 phosphorylation sites changed their phosphorylation status in a statistically significant manner, depending on the treatment and age group’ and refer to suppl fig1. Again, there is no data on the reproducibility of the results, and in my opinion the numbers of phosphopeptides identified and quantified are not very impressive posing questions on the quality of the data. With optimized phosphopeptide enrichment strategy combined with high-resolution LC-MS/MS it should be possible to quite easily identify thousands of phosphopeptides from total cell lysates.

Reply from the authors:

We agree that the amount of phosphopeptides detected is relatively low, based on our experience with cell lines or animal tissue. However, this is a direct result of the unique source of cells used here, rather than an issue with data quality or reproducibility.

The true novelty of our approach is the use of primary human cells for this type of analysis, instead of cell lines that can be propagated. This results in physiologically more meaningful results, at the tradeoff of lower cell number and therefore lower peptide abundance. Especially the use of phosphoproteomics on neonatal primary cells, isolated from the umbilical cord within 30 minutes of birth, adds a great amount of novelty in our opinion. Despite relatively low numbers of phosphopeptides, we were able to reproducibly identify changes in phosphorylation status of highly relevant proteins.

-In Suppl fig 1 the volcano plots for adult cells (Suppl fig 1B) seem highly biased, why is that?

Reply from the authors:

This is indeed the case, thank you for pointing this out. In newborn cells, we indeed observe a reduction in a proportion of phosphorylation events as compared to vehicle-treated controls, which is not observed in cells from adult study participants. This is an interesting phenomenon that was initially not discussed in the manuscript. Because treatment conditions were compared to resting conditions, we actually anticipated that the majority of changes in phosphorylation would be biased towards up-regulation, as seen in the adult cells. We hypothesize that the observed presence of down-regulated phosphorylation events observed in newborn cells may be due to the fact that birth can be an inflammatory process, and there may have been elevated levels of baseline phosphorylation under resting conditions in these cells.

We have therefore included the following statement in the manuscript (Lines 143-149):

“A reduction in phosphorylation, as compared to vehicle-treated controls, was seen for a proportion of phosphorylation events in newborn MoDCs, but not adult MoDCs. We speculate that newborn MoDCs may display an elevated level of baseline phosphorylation of certain housekeeping proteins as a result of c-section-induced activation. Analysis of these downregulated phosphorylation events did not result in significant identification of any signaling pathways (data not shown).”

-Fig 1C-D show phosphoproteomic results for two phosphopeptides, but again it is not clear what data from MaxQuant search has been used for this. Fig 1E shows ‘confirmation of MS results with western blotting’, however, the WB is not of very good quality and no fold-change values have been calculated to compare that with the MS-data.

Reply from the authors:

We have edited the text of the results section to better describe the analysis pipeline and have also included Supplementary Figure 2, showing ImageJ quantification of protein phosphorylation by Western Blot from these

blots, as well as from two additional biological replicate experiments. The text included in the manuscript reads as follows (Lines 149-161):

“Following identification of phosphorylation events that were induced by treatment in a statistically significant manner, this proportion of the data was subsequently used for pathway overrepresentation analysis using InnateDB.com. Pathway overrepresentation analysis, mining multiple protein function databases (KEGG 46, REACTOME 47, PID NCI 48, BioCarta 49, NetPath 50 and INOH 51), for increased confidence, confirmed the synergistic induction of cytokine secretion signaling pathways by R848+TDB in newborn MoDCs, as compared to stimulation with each individual agonist (Fig. 1A). This synergy is absent in adult MoDCs, where treatment with single agonists results in the greatest induction of phosphorylation events in cytokine signaling pathways (Fig. 1B). In addition, a marked synergy in phosphorylation in endocytosis-related signaling pathways was observed in newborn MoDCs, but not in adult MoDCs. Further analysis of individual proteins involved in these pathways revealed the significant phosphorylation of PKC delta playing a role in the cytokine signaling pathways as well as the endocytosis pathways, providing a possible link between these two processes.”

-Fig 1F-G (Inhibition of endocytosis with a clathrin inhibitor): for these experiments Clathrin inhibitors Pitstop@1, 2 were used. How specific are they? Pitstop2 has been shown to be a potent inhibitor of also clathrin-independent endocytosis.

Reply from the authors:

Thank you for pointing out this important detail. The data shown in Figure 1 were generated using Pitstop 1. We do see a similar reduction in TNF secretion when Pitstop 2 is used, but have not included these results in the manuscript, because pre-treatment with Pitstop 2 led to a reduction in cell viability in our hands. The mention of Pitstop 2 in the methods section was an error and has been removed.

-Fig 2A-C can only be evaluated after having more details on the phosphoproteome analysis, and Fig 2B WB images are missing sample group info.

Reply from the authors:

We have added sample and treatment group information to the WB images in Figure 2, our apologies for this omission.

As mentioned above, we have also provided Supplementary tables containing peptide intensities for all phosphopeptides detected, and have made the raw data and analysis files available through the ProteomeXchange repository.

Experiments with liposomal formulations, CD8+ T cell activation studies *in vitro* and *in vivo* newborn immunization model seem to be well done, but how well they are linked to the phosphoproteome data should be re-evaluated after having more details on the phosphoproteomics. Overall, these two parts of the manuscript are somewhat distant to each other in the current version of the manuscript.

We agree with the reviewer that there was a need to link the phosphoproteomics results with the CD8+ T cell activation studies. In order to provide a mechanistic link between the identification of cross-presentation-related phosphorylation events and the induction of CD8+ T cells, we have utilized the *in vitro* DC+CD8 coculture platform described in Figure 2D to examine the role of Sec22B, a protein that was demonstrated to be phosphorylated upon R848+TDB treatment (Figure 2A-B), and has been described by others to play a role in cross-presentation. Supplementary Figure 3 shows n=3 results from DC+CD8+ T cell co-culture experiments where the DCs were pre-treated with an siRNA pool targeting Sec22b, or with a mock siRNA pool. The siRNA pool targeting Sec22b expression reduced the protein levels of this protein by ~85%, which was enough to completely abrogate the induction of autologous antigen-specific CD8+ T cells after activation of these DCs with R848+TDB. There was no significant change in the ability of these cells to induce a CD4+ T cell response, suggesting that reduction of sec22b protein levels specifically impaired the process of antigen cross-presentation.

We once again thank the Reviewers for the opportunity to revise and improve our manuscript that is now further strengthened.

REVIEWER COMMENTS

Reviewer #1 (Remarks to the Author):

The changes made to the manuscript have certainly improved it. I have a couple of further minor points:

1. Can you check the numbering of the Figures / supplementary figures as there seems occasion where they are incorrect e.g. the second to last line of the Liposome formulation section of the methods section refers to "The conformation of pre-F after incorporation into DDA liposome formulations was evaluated by binding to monoclonal antibodies AM-14, 5C4 and Motavizumab by dot blot (Supplementary Fig.2)." but I believe this should refer to Supp Fig. 4.
2. In Supp Fig 2. I am not clear what the x-axis refers to or the different colours of the bars. Also the bottom right graph y-axis doesn't start at 0 like all the others.

Reviewer #2 (Remarks to the Author):

It is still not clear which file from MQ combined was used for the data analysis in Perseus; 'evidence' does not have the info on the sample level. The new supplementary tables provided are not very useful, they only show phosphopeptide intensities after log₂ transformation, and the statistically significant differences after Perseus analysis are not shown in the tables or as part of PRIDE submission. Further, the Suppl Table having the adult data only has data from four individuals, not five like stated in the manuscript. The same is true for the data uploaded into PRIDE.

I uploaded the adult data into Perseus and with paired t-test using the criteria written in mat+met (In Perseus, peptide intensities were expressed as fold change over vehicle-treated samples, log₂-transformed and z-score normalized within each sample to control for inter-sample variation. A weighted p-value cutoff was applied (s₀=2, FDR=0.05) to determine statistically significant changes in phosphorylation over vehicle-treated conditions.) did not get any statistically significant hits after z-score normalization. For the non-normalized peptide intensity data provided in the suppl table I get statistically significant differences but only for the comparison TDB_vehicle.

It is clear that based on PCA there are large individual differences how the donors respond to different treatments. This can be either real biological differences or artefacts coming from e.g phosphopeptide enrichment step or other parts of the phosphoproteome workflow. Also, the general downregulation in phosphorylation upon treatment seen in newborns poses questions re phosphoproteome data quality/reproducibility, the new discussion included on that is highly speculative and there is no data to proof that.

The authors state in the rebuttal that 'Despite relatively low numbers of phosphopeptides, we were able to reproducibly identify changes in phosphorylation status of highly relevant proteins.' Again, there is no data showing the reproducibility of the results; based on PCA there is no clustering of the samples based on different treatments.

Overall I think the authors have not addressed my previous concerns very carefully and the phosphoproteome part of the manuscript is still lacking a lot of details and I have serious concerns on the quality of that data.

Dear Reviewers,

We sincerely thank you for your consideration of our revised manuscript entitled "A single immunization with CAF08 provides newborns with Th1-mediated protection against Respiratory Syncytial Virus infection" (Manuscript ID: NCOMMS-21-03801A).

We were pleased to learn that Reviewer 1 viewed our manuscript as improved by our prior revisions and have now also fully addressed their remaining minor points.

We thank Reviewer 2 for their very careful reading of our manuscript and have taken their concerns very seriously. Accordingly, we have subjected our entire proteomics dataset to complete re-analysis by experts within Dr Hanno Steen's Laboratory and have made substantial revisions to the manuscript to improve both the transparency of our approach to the data analysis and quality and reproducibility of our results.

We greatly appreciate the expert feedback provided to us by both reviewers and we have now fully addressed their feedback. As outlined below, we have responded to each of the reviewer comments, and have indicated for each point where and how the manuscript was changed.

A revised manuscript indicating all changes made using 'track changes' has been provided with this re-submission, as well as a 'clean' version.

RESPONSE TO REVIEWER COMMENTS:

Reviewer #1 (Remarks to the Author):

The changes made to the manuscript have certainly improved it. I have a couple of further minor points:

1. Can you check the numbering of the Figures / supplementary figures as there seems occasion where they are incorrect e.g. the second to last line of the Liposome formulation section of the methods section refers to "The conformation of pre-F after incorporation into DDA liposome formulations was evaluated by binding to monoclonal antibodies AM-14, 5C4 and Motavizumab by dot blot (Supplementary Fig.2)." but I believe this should refer to Supp Fig. 4.

2. In Supp Fig 2. I am not clear what the x-axis refers to or the different colours of the bars. Also the bottom right graph y-axis doesn't start at 0 like all the others.

Reply from the authors:

We would like to thank Reviewer 1 for their expertise and for picking up on these important points.

1. Per the Reviewer's suggestion, we have corrected the manuscript on line 452, where indeed the text should refer to Supplementary Figure 4, and not Supplementary Figure 2. In addition, we have double-checked that all Figures are correctly numbered and all text refers to the correct Figure throughout the manuscript.

2. To address the Reviewer's point, we have changed Supplementary Figure 2 accordingly. x-axes have been labeled with "stimulation time (minutes), bar colors have been annotated with their respective treatment conditions, and the y-axis of the bottom-right graph now starts at 0.

Reviewer #2 (Remarks to the Author):

1. It is still not clear which file from MQ combined was used for the data analysis in Perseus; 'evidence' does not have the info on the sample level. The new supplementary tables provided are not very useful, they only show phosphopeptide intensities after log2 transformation, and the statistically significant differences after Perseus analysis are not shown in the tables or as part of PRIDE submission Further, the Suppl Table having the adult data only has data from four individuals, not five like stated in the manuscript. The same is true for the data uploaded into PRIDE.

Reply from the authors:

We would like to thank Reviewer 2 for taking the time to make sure all data are represented in a clear and reproducible manner. We take quality and reproducibility of our findings very seriously. With the help of feedback from Reviewer 2, we have completely re-analyzed all our phospho-proteomic data and have now made sure that our analysis pipeline is both transparent and reproducible.

- Supplementary Figure 1 shows the analysis pipeline as exported from the Perseus software.
- Supplementary Table 1 is the actual output generated by Maxquant software.
- Supplementary Tables 2-10 were generated (as shown in the pipeline figure in Supplementary Figure 1) with each individual analysis step (e.g., transformation, imputation, volcano graph, output tables), so that reviewers and readers can follow each step of the analysis and how the final result was generated.

To clarify, the adult proteomic validation cohort (this manuscript focuses on the newborns, which is why the n is higher for that group) was indeed N = 4, as stated throughout the manuscript. The only mention of N = 5 adults is for Figure 2D, which is not derived from a proteomic dataset, but rather from an *in vitro* cell culture validation experiment where indeed cells from N = 5 adults were used.

2. I uploaded the adult data into Perseus and with paired t-test using the criteria written in mat+met (In Perseus, peptide intensities were expressed as fold change over vehicle-treated samples, log₂-transformed and z-score normalized within each sample to control for inter-sample variation. A weighted p-value cutoff was applied (s₀=2, FDR=0.05) to determine statistically significant changes in phosphorylation over vehicle-treated conditions.) did not get any statistically significant hits after z-score normalization. For the non-normalized peptide intensity data provided in the suppl table I get statistically significant differences but only for the comparison TDB_vehicle.

Reply from the authors:

Our re-analysis of the data has provided us with an indication of how this apparent discrepancy described by the Reviewer could have happened.

As shown in Supplementary Figure 1, and undoubtedly also observed by Reviewer 2 when analyzing our publicly repositied data, each volcano graph showing fold changes over vehicle-treated samples has a large number of peptides that are > 1.3 on the y-axis, meaning they have a p-value < 0.05.

Indeed, we also found that when the FDR- and permutation-based parabolic cutoff lines are applied in Perseus, as described in the previous version of the Methods section, these lines do not always include the same significant phosphopeptides as we originally reported. In fact, we found that the placement of these lines was entirely dependent on which release version of the Perseus software was used, and in some cases resulted in placement of these lines such that it could not allow for phosphopeptides to be considered significant unless a p-value lower than 0.000001 was achieved.

To ensure that there is no ambiguity in the interpretation of our data, and to allow all readers to evaluate our data independently of whether they are familiar with Perseus software, we have chosen to apply a unweighted, universally recognized constant significance cutoff of p<0.05, as shown in the updated volcano graphs in Supplementary Figure 1. Peptides considered statistically significant are also clearly indicated in Supplementary Tables 5-10. This re-analysis has minimally affected the pathway overrepresentation analysis (Figure 1A), and did not affect the reported intensities of individual phosphopeptides shown in Figures 1+2.

We do recognize that this analysis comes at the potential risk of increasing the chances of false discovery. We agree with Reviewer 2 that data reproducibility, but also controlling for false discovery, is extremely important. We have therefore employed the most rigorous form of control for false discovery: through experimental validation. All phospho-proteomic events that we describe as relevant in Figures 1 and 2 were experimentally validated in the following ways:

1. Individual phosphorylation events were confirmed by western blotting in 3 'new'/different study participants, either using phosphorylation-specific antibodies (Figure 1 E-G, Supplementary Figure 2), or using phos-tag gels (Figure 2B).

2. The roles of reported phosphorylation events on cellular response to stimulation were confirmed using either small-molecule inhibitors blocking the function of the phosphorylated protein (Figure 1 F,G,H), or using an siRNA targeting the phosphorylated protein (Supplementary Figure 3).

3. Physiological relevance of the biological pathways reported were confirmed using functional assays demonstrating that these pathways were indeed induced in the cells and are relevant to immune activation (Figure 2D, Supplementary Figure 3B,C).

4. The induction of antigen cross-presentation, an immune pathway we hadn't considered relevant until it was identified by our phospho-proteomic data, was confirmed in an animal model, as evidenced by the induction of antigen-specific CD8⁺ T cells in newborn mice immunized with a CAF08-adjuvanted protein antigen (Figure 4D).

We have now clarified these points, together with a description of the limitations of our approach, in the Discussion, on lines 376-388: "Our study also has limitations. As expected, human primary cells have more diverse signatures than uniform cell lines, resulting in donor-specific variability observed in PCA as well as volcano graphs. This highlights the importance of controlling for false discovery. Potential false discovery within the phosphoproteomic observations was controlled for via extensive validation of individual phosphorylation events by western blotting, as well as confirmation of immune signatures via independent immunologic assays employing leukocytes from additional/distinct study participants *in vitro* and in mice *in vivo*."

Overall, these revisions have substantially enhanced the rigor and clarity of our manuscript.

3. It is clear that based on PCA there are large individual differences how the donors respond to different treatments. This can be either real biological differences or artefacts coming from e.g phosphopeptide enrichment step or other parts of the phosphoproteome workflow. Also, the general downregulation in phosphorylation upon treatment seen in newborns poses questions re phosphoproteome data quality/reproducibility, the new discussion included on that is highly speculative and there is no data to proof that. The authors state in the rebuttal that 'Despite relatively low numbers of phosphopeptides, we were able to reproducibly identify changes in phosphorylation status of highly relevant proteins.' Again, there is no data showing the reproducibility of the results; based on PCA there is no clustering of the samples based on different treatments. Overall I think the authors have not addressed my previous concerns very carefully and the phosphoproteome part of the manuscript is still lacking a lot of details and I have serious concerns on the quality of that data.

Reply from the authors

We appreciate the Reviewer's concerns with quality and reproducibility of our data. We have gone to great lengths to ensure that we are reporting our results and analysis methods in a transparent way and have made substantial revisions to the manuscript to improve clarity and uniformity of our analysis. We have also taken extensive measures to control for false discovery, by validating key phosphorylation events and relevant signaling pathways using additional *in vitro* and *in vivo* experiments (Figure 1 E-H, Figure 2 B-D, Figure 4D, Supplementary Figures 1-3). In summary, our proteomics data have been useful in generating hypotheses, which were further tested and validated in this manuscript through rigorous experimentation.

Our previous statement in the discussion regarding downregulation of phosphorylation events in newborn samples was not interpreted as definitive and, as the reviewer points out, was speculative. To clarify this, we have amended this section (lines 147-148-153).

We also agree with the reviewer that there are large differences in how the donors respond to different treatments. To maximize the likelihood that our results were relevant to human newborns *in vivo*, we studied human neonatal primary leukocytes rather than uniform cell lines. Accordingly, consistent with prior studies of human neonatal cord and adult blood mononuclear cells in conventional immunological experiments (van Haren et al., *J. Immunol.* 2016; van Haren et al., *Cytokine* 2016; Levy et al., *J. Immunol.* 2006; Scheid et al., *Front. Immunol.* 2018), as well as in proteomic experiments (Bennike et al., *Front. Immunol.* 2020; Lee et al., *Nat. Commun.* 2019; Oh et al. *Mol. Cell. Prot.*, 2016), and in phospho-proteomic experiments (Barrachina et al., *Art. Thromb. Vasc. Biology.* 2020; van den Biggelaar et al., *Blood.* 2014; Izquierdo et al., *Thromb. Haemost.* 2020; Welz et al., *Int. J. Mol.*

Sci. 2019), the quantity and distribution of phosphopeptides is expected to be more diverse than in proteomic studies that employ cell lines.

We once again thank the Reviewers for the opportunity to revise and improve our manuscript that is now further strengthened.

We very much hope it is now acceptable for publication in *Nature Communications*.

REVIEWER COMMENTS

Reviewer #2 (Remarks to the Author):

I still disagree with the authors on the processing of the phosphoproteomics data; I think starting from evidence.txt is not correct and they have also not done proper data filtering and evaluation of the quality of the data (data is not filtered to include reproducible identifications/quantifications and not for phosphorylation site probability, also no evaluation how much of the values used for t-test are based on imputation etc).

I think the authors should go through MaxQuant/Perseus tutorials, especially <https://www.youtube.com/watch?v=52c8YDrcZdc> and if they disagree on the data processing with the developers of the softwares give very thorough answers why this is.

I re-processed the adult data in Perseus ver 1.6.15.0 using the phospho(STY).txt file from adult samples. Data processing followed the instructions given in Perseus http://coxdocs.org/doku.php?id=perseus:user:use_cases:modifications

The log2 transformed data was filtered to include only those intensities which can be found at least in 2/4 replicates in at least one group (this could have been even more stringent requiring at least 75% identifications in at least one group), and with phosphorylation site probability >0.75, and missing values were imputed from normal distribution. Histograms show that in many samples most values are based on imputation. All these should have been removed before further analysis (outliers).

So again, I think the data needs to be re-analyzed and the quality evaluated, and outliers removed. I do understand that the starting material is much more heterogenous than working with cell lines, but if the authors do not have good quality phosphoproteome data from most of the samples used for the analysis maybe they should aim in less quantitative approach in the data analysis and reporting? This then poses the question whether the phosphoproteome data should be published in Nature Communications.

Dear Reviewers,

We once again thank you for your latest consideration of our revised manuscript entitled "A single immunization with CAF08 provides newborns with Th1-mediated protection against Respiratory Syncytial Virus infection" (Manuscript ID: NCOMMS-21-03801C).

We were pleased to learn that Reviewer 1 no longer expresses any concerns with the publication of our manuscript. We thank Reviewer 2 for their continued diligence in ensuring that the mass-spectrometry data are analyzed in the most rigorous manner. We are pleased to report that we have re-analyzed the mass-spectrometry exactly as Reviewer 2 proposed in their latest comments and note that our initial observations remain valid and that the key observations, which were also confirmed through additional experimentation with immunological readouts, remain statistically significant using the analysis method recommended by Reviewer 2.

We greatly appreciate the expert feedback provided to us by both reviewers and we have now fully addressed their feedback. As outlined below, we have responded to each of the reviewer comments, and have indicated for each point where and how the manuscript was changed.

A revised manuscript indicating all changes made using 'track changes' has been provided with this re-submission, as well as a 'clean' version.

RESPONSE TO REVIEWER COMMENTS:

Reviewer #2 (Remarks to the Author):

- "I still disagree with the authors on the processing of the phosphoproteomics data; I think starting from evidence.txt is not correct and they have also not done proper data filtering and evaluation of the quality of the data (data is not filtered to include reproducible identifications/quantifications and not for phosphorylation site probability, also no evaluation how much of the values used for t-test are based on imputation etc)."

Reply from the authors:

We thank the reviewer for their helpful input on these analyses. In our revised manuscript, we have re-analyzed the mass-spectrometry data as suggested, starting with the phospho(STY).txt file. We have also included data filtering for reproducible identifications/quantifications, as well as phosphorylation site probability, and have excluded a few samples in which > 50% of the data were imputed values. The revised Supplementary Figure 1 has an updated illustration of the new analysis pipeline as exported from the Perseus software, and Supplementary Tables 1-6 contain the actual output generated by Perseus software. The Methods section of the manuscript has been modified in Lines 508-520 to reflect this new analysis method. Pathway analysis and intensities of individual peptides reported in Figures 1 and 2 have been updated accordingly. Of note, employing the analysis method recommended, main observations in Figures 1 and 2 remain statistically significant.

- "I think the authors should good through MaxQuant/Perseus tutorials, especially <https://www.youtube.com/watch?v=52c8YDrcZdc> and if they disagree on the data processing with the developers of the softwares give very thorough answers why this is."

Reply from the authors:

Thank you for this helpful information. We have used the instructions here as well as in the link provided in the next comment to guide us through the analysis. To clarify, we did not disagree with the developers of the

Perseus Software on how data should be processed. We simply noted that we observed differences in analysis outcome depending on which release version of the Perseus software was used.

-“I re-processed the adult data in Perseus ver 1.6.15.0 using the phospho(STY).txt file from adult samples. Data processing followed the instructions given in Perseus http://coxdocs.org/doku.php?id=perseus:user:use_cases:modifications. The log2 transformed data was filtered to include only those intensities which can be found at least in 2/4 replicates in at least one group (this could have been even more stringent requiring at least 75% identifications in at least one group), and with phosphorylation site probability >0.75, and missing values were imputed from normal distribution. Histograms show that in many samples most values are based on imputation. All these should have been removed before further analysis (outliers).”

Reply from the authors:

We thank the Reviewer for this detailed re-analysis. We re-analyzed our data using these exact parameters as well, filtering to include only those intensities which can be found at least in 2/4 biological replicates in at least one group for the Adult dataset, and at least in 5/9 biological replicates in the Newborn group. Phosphorylation site probability was set to >0.75, and missing values were imputed from normal distribution.

Similar to the Reviewer, we also noted that there was a subset of replicates where the majority of the intensities were imputed values. Most of the 52 samples (36 newborn samples, 16 adult samples) had < 25% imputed values, but there were a few samples that instead had > 80% imputed values. These samples (<25% valid values) were removed from subsequent analysis per the Reviewer’s recommendation. We thank the Reviewer for this observation because exclusion of these replicates has increased the quality of the data as observed in Supplementary Figure 1. The Methods section of the manuscript has been modified in Lines 508-520 to reflect this information.

-“So again, I think the data needs to be re-analyzed and the quality evaluated, and outliers removed. I do understand that the starting material is much more heterogenous than working with cell lines, but if the authors do not have good quality phosphoproteome data from most of the samples used for the analysis maybe they should aim in less quantitative approach in the data analysis and reporting? This then poses the question whether the phosphoproteome data should be published in Nature Communications.”

Reply from the authors:

We appreciate the Reviewers’ focus on scientific rigor and their appreciation for the importance of studying primary age-specific human cells and the inherent heterogeneity that comes with their use as opposed to cell lines, and the complexity that comes with that.

The fact that after rigorous re-analysis, per Reviewer 2’s suggestion, the original observations described in our manuscript with respect to identified pathways and peptides remain significant further demonstrates the quality of our study and validity of our conclusions. Moreover, as mentioned in prior revisions, we intentionally used the phospho-proteomic dataset as a hypothesis-generating tool, with all observations subsequently confirmed via extensive experimentation including confirmation of phosphorylation events by Western Blotting in 3 new biological replicates, siRNA- and small-molecule inhibitor- inhibition of key phosphorylation events and biological pathways, and functional cell biology readouts that all confirmed our original observations in an age-specific manner.

Following the latest recommendations of Reviewer 2 with respect to re-analysis of the mass-spectrometry data has substantially improved our data analysis and reporting.

We once again thank the Reviewers for the opportunity to revise and improve our manuscript that is now further strengthened. Given the increasing interest in pediatric immunization against viral infections, and the potentially broad applicability of adjuvantation systems such as CAF08 not only against RSV but also for pediatric vaccines against other viral pathogens such as coronaviruses, we are keen to receive a timely, and hopefully positive, decision.

We very much hope our robust manuscript is now acceptable for publication in *Nature Communications*.

Sincerely,

Simon van Haren, Ph.D.

REVIEWER COMMENTS

Reviewer #2 (Remarks to the Author):

The authors have done complete re-processing of the phosphoproteome data according to suggestions from previous review, but there are still some discrepancies in the results/ supplementary tables that should be corrected.

Suppl tables 1-3 (adult data): columns X-AA show t-test results for the different comparisons, and columns AJ-AR have the p-values and difference-values for these. Here, t-test is done with p-value criteria only, but mat+met/results text state t-test was done with perm based FDR. Is that shown in column A? No p- and q-values for that test are shown. Also, after filtering the V group only has two samples left so the statistics is not very impressive, that should at least be commented in the discussion.

Suppl tables 4-6 (newborn data) are in different format than suppl tables 1-3, why? Would be easier for the readers to have them in the same format. Here there is also something wrong with filtering based on localization score; there are several peptides with localization score probability <0.75, please check the filtering and correct the tables.

Dear Reviewers,

We once again thank you for your latest consideration of our revised manuscript entitled "A single immunization with CAF08 provides newborns with Th1-mediated protection against Respiratory Syncytial Virus infection" (Manuscript ID: NCOMMS-21-03801C).

We thank Reviewer 2 for the detailed consideration of our mass spectrometry analysis and were pleased to learn that only minor points of concern remained after our re-analysis of the data. Per the Reviewer's recommendations, we have addressed each of the remaining minor concerns with edits to the manuscript text and changes to the supplementary tables. Below we outline point-by-point how we have addressed each comment.

A revised manuscript indicating all changes made using 'track changes' has been provided with this re-submission, as well as a 'clean' version.

RESPONSE TO REVIEWER COMMENTS:

Reviewer #2 (Remarks to the Author):

-*"The authors have done complete re-processing of the phosphoproteome data according to suggestions from previous review, but there are still some discrepancies in the results/ supplementary tables that should be corrected."*

Reply from the authors:

Thank you for identifying these discrepancies. We have corrected them as you suggested, see below.

-*"Suppl tables 1-3 (adult data): columns X-AA show t-test results for the different comparisons, and columns AJ-AR have the p-values and difference-values for these. Here, t-test is done with p-value criteria only, but mat+met/results text state t-test was done with perm based FDR. Is that shown in column A? No p- and q-values for that test are shown. Also, after filtering the V group only has two samples left so the statistics is not very impressive, that should at least be commented in the discussion."*

Reply from the authors:

Indeed, the t-tests were done with permutation-based FDR as described in the Materials and Methods. The (now removed) columns X-AA and AJ-AR in Supplementary Tables 1-3 were left behind from a prior t-test that was not used, as it was lacking the additional stringency of the permutation-based FDR. These have now been removed. Columns A, B, and C reflect the statistical significance if achieved by t-test with permutation-based FDR, the $-\log_{10}$ p-value, and the fold-change over vehicle, respectively.

We also acknowledge that, although the latest re-analysis of the phosphoproteomic data was a significant improvement from the prior analysis, filtering of replicates containing mostly imputed data has resulted in the control cohort (Adult) having only 2 replicates in the vehicle control group. Per the Reviewer's suggestion, we have acknowledged this in the Discussion of the manuscript, see lines 384-390 (changes highlighted manuscript).

-*"Suppl tables 4-6 (newborn data) are in different format than suppl tables 1-3, why? Would be easier for the readers to have them in the same format. Here there is also something wrong with filtering based on localization score; there are several peptides with localization score probability <0.75, please check the filtering and correct the tables."*

Reply from the authors:

We have made Supplementary Tables 4-6 into the same format as Supplementary Tables 1-3. We have also fixed the issue where peptides with a localization probability of <0.75 were not all filtered out. Per the Reviewer's feedback, any such peptide has now been filtered out.

We once again thank the Editor and Reviewers for the opportunity to revise and further improve our manuscript that is now further strengthened. Given the increasing interest in pediatric immunization against viral infections, and the potentially broad applicability of adjuvantation systems such as CAF08 not only against RSV but also for pediatric vaccines against other viral pathogens such as coronaviruses, we are keen to receive a timely, and hopefully positive, decision.

We very much hope our robust study is now acceptable for publication in *Nature Communications*.

Sincerely,

Simon van Haren, Ph.D.

REVIEWER COMMENTS

Reviewer #2 (Remarks to the Author):

i think the authors have addressed all my final concerns from the last review round.

Dear Reviewers

We once again thank you for your latest consideration of our revised manuscript entitled " *CAF08 adjuvant enables single dose protection against Respiratory Syncytial Virus infection in murine newborns*" (Manuscript ID: NCOMMS-21-03801D).

We were pleased to Learn that both peer reviewers felt that their concerns have now been addressed. We would like to thank both peer reviewers for their careful consideration and attention to detail.

RESPONSE TO REVIEWER COMMENTS:

Reviewer #2 (Remarks to the Author):

-“i think the authors have addressed all my final concerns from the last review round.”

Reply from the authors:

Thank you for this assessment, we are pleased that there are no remaining concerns.

We once again thank the Reviewers for the opportunity to revise and further improve our manuscript. Given the increasing interest in pediatric immunization against viral infections, and the potentially broad applicability of CAF08 not only against RSV but also for pediatric vaccines against other viral pathogens such as coronaviruses, we are keen to receive a timely, and hopefully positive, decision.

Sincerely,

Simon van Haren, Ph.D.